# Nociceptive withdrawal reflexes of the trunk muscles in chronic low back pain

**Hugo Massé-Alarie**[1,2]*, **Genevieve V. Hamer**[1], **Sauro E. Salomoni**[1], **Paul W. Hodges**[1]

**1** The University of Queensland, NHMRC Centre of Clinical Research Excellence in Spinal Pain, Injury & Health, School of Health & Rehabilitation Sciences, Brisbane, Qld, Australia, **2** Université Laval, *Cirris, CIUSSS-Capitale Nationale*, Quebec City, Qc, Canada

* hugo.masse-alarie@fmed.ulaval.ca

**Data Availability Statement:** The anonymised dataset is available at https://doi.org/10.5683/SP3/OVTYIN, an institutional repository (Scholar Dataverse).

## Abstract

Individuals with chronic low back pain (CLBP) move their spine differently. Changes in brain motor areas have been observed and suggested as a mechanism underlying spine movement alteration. Nociceptive withdrawal reflex (NWR) might be used to test spinal networks involved in trunk protection and to highlight reorganization. This study aimed to determine whether the organization and excitability of the trunk NWR are modified in CLBP. We hypothesized that individuals with CLBP would have modified NWR patterns and lower NWR thresholds. Noxious electrical stimuli were delivered over S1, L3 and T12, and the 8th Rib to elicit NWR in 12 individuals with and 13 individuals without CLBP. EMG amplitude and occurrence of *lumbar multifidus* (LM), *thoracic erector spinae*, *rectus abdominus*, *obliquus internus* and *obliquus externus* motor responses were recorded using surface electrodes. Two different patterns of responses to noxious stimuli were identified in CLBP compared to controls: (i) abdominal muscle NWR responses were generally more frequent following 8th rib stimulation and (ii) occurrence of erector spinae NWR was less frequent. In addition, we observed a subgroup of participants with very high NWR threshold in conjunction with the larger abdominal muscle responses. These results suggest sensitization of NWR is not present in all individuals with CLBP, and a modified organization in the spinal networks controlling the trunk muscles that might explain some changes in spine motor control observed in CLBP.

## Introduction

Individuals with chronic low back pain (CLBP) move differently to painfree controls. For example, studies have reported increased activation of superficial erector spinae [1] and abdominal [2] muscles during forward bending, and delayed activation of deep multifidus muscles during postural tasks [3] and walking [4] in participants with CLBP compared to painfree controls. Differences in spine motor control have been observed at multiple levels of the motor system [5] including differences in function/organization of the trunk-muscle representations within the primary motor cortex (M1) [6–10] and relate to less optimal spine control [10,11]. Neural processing in supplementary motor areas [12] and functional connectivity in the cerebellum [13,14] which are involved in motor control [15,16], also differ in CLBP

**Funding:** This study was funded by a Program Grant from the National Health and Medical Research Council of Australia (NHMRC) of Australia (APP1091302). HMA was supported by a Postdoctoral Fellowship from the Canadian Institutes for Health Research (358797) and is supported by a Research Scholar Award from Fonds de recherche du Québec – Santé (HMA: 281961). P.H. is supported by a Fellowship (APP1194937) from the NHMRC. The funders had no role in study design, data collection and analysis, decision to publish, or preparation of the manuscript. None of the authors have potential conflicts of interest to be disclosed.

**Competing interests:** The authors have declared that no competing interests exist.

compared to painfree controls. The assessment of spinal motor networks that control trunk muscles is challenging.

The spinal motor network of the trunk was tested in one study using noxious skin stimulation [17] or using stretch reflex responses to mechanical tap on paravertebral muscle [18–20]. Cutaneous low back pain did not influence the short-latency response [17,20] but reduced the long-latency response of the stretch reflex [17]. This suggests cutaneous pain may affect supraspinal but not spinal excitability. The nociceptive withdrawal reflex (NWR) also provides information about spinal cord networks. The NWR is a protective reflex with two components: a stereotypical early response controlled at the spinal cord and a late response that adapts to the task context and is influenced by supraspinal control [21–25]. Recent work suggests fine tuning of the trunk muscle NWR–noxious stimulation elicited site-specific patterns of muscle response ('*motor strategy*') that were functionally relevant to withdraw the trunk from the noxious stimulus, and the response of each muscle depended on the stimulation site ('*receptive field*') in a manner consistent with its biomechanical function [26]. Trunk NWRs may provide insight into differences in the function of the neural networks controlling the trunk muscles at spinal and supraspinal levels.

Studies of CLBP [27,28] and other chronic painful conditions have reported hyperexcitability of the NWR to foot stimulation, which is interpreted as central sensitization [29,30]. This stimulation site is distant from the painful region and might inform about the general excitability of pain processing but does not provide insight into potential changes in spinal/supraspinal control of muscles in the painful region. Although studies have tested sensitivity to sensory stimuli applied to the lumbar spine in CLBP [31], none has investigated the NWR of the trunk to probe the excitability of networks involved in pain or trunk motor control. We hypothesized that CLBP would involve a lower threshold to NWR at the back.

Changes in trunk motor control in pain are variable, with evidence of both generalized increase [32] and compromised activity (e.g., lumbar multifidus [3]). These changes are linked with differences in the organization in the higher central nervous system regions [10]. We hypothesized that CLBP would be characterized by either less specificity of trunk muscle NWR receptive field (similar muscle activation across stimulation sites) or greater/lesser activation of some muscles (modified motor strategy).

This exploratory study compared NWR in CLBP with data reported previously for painfree individuals [26]. The aims were to test whether: (i) NWR excitability (stimuli at different trunk sites), differs between individuals with and without CLBP; and (ii) the NWR organization of the trunk (receptive field and motor strategy) differed between groups.

## Materials & methods

### Study design and participants

This study compares data for a group of individuals with CLBP with previously published data of NWR in painfree individuals [26]. In the absence of previous data on NWR of the trunk muscles in CLBP, this study was designed as an exploratory study using a sample of twelve individuals with CLBP recruited from the local community through advertisements. Considering the difficulty to recruit participants with CLBP in the study (noxious stimuli elicited high intensity of pain), the sample size was based on feasibility. Descriptive characteristics are detailed in Table 1. Participants with CLBP were included if they were: aged between 18 and 60 years, had LBP for at least 3 days/week for at least 3 months, and reported low back pain during the preceding week with an average intensity of >2 and <6 on a 10-cm visual analog scale (VAS; anchored with "no pain" at 0 cm and "worst pain imaginable" at 10 cm). We considered it unlikely that individuals with CLBP higher than 6/10 would tolerate the experiment.

**Table 1. Participant characteristics (mean (SD)).**

| Characteristic | LBP (n = 12) | CTL (n = 13) | p | Low-threshold LBP (n = 5) | High-threshold LBP(n = 7) | p |
|---|---|---|---|---|---|---|
| Age (years) | 29 (10) | 30 (11) | 0.85 | 29 (13) | 28 (7) | 0.80 |
| Gender (M: F) | 5: 7 | 7: 6 | 0.54 | 2: 5 | 3: 2 | 0.57[1] |
| Height (cm) | 169 (9) | 170 (12) | 0.98 | 169 (10) | 170 (9) | 0.99 |
| Weight (kg) | 63 (13) | 66 (11) | 0.49 | 63 (12) | 62 (16) | 0.72 |
| BMI (kg/m) | 21.6 (3.0) | 23.0 (2.1) | 0.13 | 21.9 (2.6) | 21.2 (3.8) | 0.28 |
| Laterality (R: L) | 9: 3 | 12: 1 | 0.32[1] | 5: 2 | 4: 1 | 0.27[1] |
| Pain duration (months) | 81 (106) | - | | 89 (109) | 71.4 (114.9) | 0.76 |
| Average pain* (/10) | 4.5 (1.6) | - | | 4.6 (1.8) | 4.4 (1.5) | 0.88 |
| Current pain (/10) | 2.8 (1.6) | - | | 3.1 (1.7) | 2.4 (1.7) | 0.64 |
| Highest pain (/10) | 7.1 (2.0) | - | | 7.6 (1.7) | 6.4 (2.3) | 0.43 |
| ODI (0–100%) | 18.0 (10.5) | - | | 21.5 (11.7) | 13.1 (6.8) | 0.15 |
| PCS (0–52) | 16.7 (8.5) | - | | 16.1 (8.1) | 17.4 ± 9.9) | 1.00 |
| TSK (17–68) | 37.2 (6.4) | - | | 36.9 (7.6) | 37.6 (5.1) | 1.00 |
| painDETECT (0–38) | 8.3 (3.7) | - | | 7.3 (4.2) | 9.8 (2.5) | 0.27 |
| CSI (0–100) | 26.8 (7.5) | - | | 28.6 (7.3) | 24.2 (7.9) | 0.34 |

SD: Standard deviation; BMI: Body Mass Index; VAS: Visual Analog Scale; ODI: Oswestry Disability Index; PCS: Pain Catastrophizing Scale; TSK: Tampa Scale of Kinesiophobia; CSI: Central Sensitisation Inventory.

[1]Fisher Exact test used.

*Average pain = average pain reported for the two previous weeks.

Exclusion criteria included; the presence of radicular pain (leg pain below the knee with numbness and/or pins and needles), numbness in the lower back, history of spinal or lumbar surgery, major neurological disorders, major medical diseases (e.g., cardiovascular disease, diabetes mellitus), psychiatric illness, insufficient knowledge of the English language to understand instructions and project description, pregnancy in the preceding 12 months, body mass index (BMI) >30kg/m$^2$, intake of opioids and/or antidepressants within 2 weeks prior to testing, and intake of other analgesics within 48 hours before testing.

The control group (CTL) was 14 pain-free participants who had been recruited for a previous study of the NWR of the trunk muscles [26]. From that sample, one participant with a BMI >30kg/m$^2$ was excluded. In addition to the criteria described above, participants were also excluded if they had; any LBP at the time of testing or that necessitated treatment in the last year or a history of chronic pain in any region of the body. Participants provided written informed consent. The local Human Research Ethics Committee of the University of Queensland approved the study (#2004000654) in accordance with the Declaration of the World Medical Association (www.wma.net).

## Questionnaires for pain and psychosocial features

Participants in the CLBP group completed questionnaires to assess pain, disability and psychosocial features. These were: Oswestry disability index (ODI), Tampa Scale of Kinesiophobia, painDETECT, Pain Catastrophizing Scale and Central Sensitization Inventory.

## Electromyography (EMG)

Pairs of surface electrodes (Ag/AgCl disks, 10 mm in diameter, Blue Sensor N, Ambu, Denmark) were placed on the right side over superficial lumbar *multifidus* ([LM] - 2 cm lateral to

L5 spinous process), thoracic *erector spinae* ([TES]– 4–5 cm lateral to T12 spinous process), *rectus abdominis* (RA), *obliquus externus* (OE) and *internus* (OI) *abdominis* muscles [33]. A ground electrode was placed over the right iliac crest. SENIAM recommendations were followed for skin preparation and electrode placement [34]. EMG data were amplified 1000 times, band-pass filtered between 5 Hz and 1 kHz (NL125 Neurolog EMG amplifier, Digitimer Ltd, Hertfordshire, United Kingdom) and sampled at 2 kHz using a Power1401 Data Acquisition System with Spike2 software (Cambridge Electronic Design, Cambridge, United Kingdom).

### Noxious electrical stimulation

Electrical stimuli were delivered as a constant current pulse train of 5 single 0.2 ms square-wave pulses with a 2-ms inter-pulse interval (Digitimer DS7AH [maximal current 1A], Hertfordshire, United Kingdom) using a 2-branch handheld 'probe' (gold electrodes, 10 mm inter-electrode distance, ~0.8 mm$^2$ contact). The experimenter positioned the probe on the participant's skin and carefully monitored its position and the applied pressure during each block of stimulation. This was perceived by the participant as a single stimulus. In separate trials, stimuli were applied over the sacrum (spinous process of S1), the spinous processes of L3 and T12, and the lateral side of the right 8$^{th}$ rib (Rib). To determine the NWR threshold, the current intensity began at 2 mA with the probe placed at L3. This was increased by 2-mA increments until a response was evoked in the LM EMG recording within the 40-200-ms window post-stimulation [35]. In the case of participants with high threshold, 5-mA increment was used. Although this was generally characterized by excitation, inhibition was sometimes observed as the earliest response when the participant was sitting. Motor response was identified visually when EMG clearly increases/decreases in comparisons to the background EMG signal (i.e., prior to stimulation). Visual identification from an expert is the gold standard to identify evoked responses [36]. At this point, the stimulus intensity was slightly increased/decreased until LM responses were evoked in 50% of the stimulations (3 out of 6). Then, the latter steps were repeated until the intensity was confirmed and this intensity was considered the NWR threshold. If no NWR was detected in the LM at 100 mA, TES was used to determine the threshold (2 of 25 participants). This method allows to determine an intensity to stimulate other trunk sites to estimate motor strategies and receptive fields (aim #2).

The intensity of stimulation used for the main trial was 2 times the NWR threshold. The level of pain induced by the stimulation at 2 times NWR over the L3 spinous process was rated on an 11-point numerical rating scale (NRS; anchored with "no pain" at 0, and "worst pain imaginable" at 10). NWR threshold was only tested at L3 using LM motor responses. Considering the duration taken to find the NWR threshold, it was not feasible to identify thresholds individually at each site and in both positions. To enable comparisons between sites, the stimulation intensity at other sites was adjusted to elicit the same level of pain as stimulation at 2x NWR threshold at L3. At sites other than L3, participants rated the pain intensity following the first stimulus. If pain intensity differed from the target by 2 or more out of 10 (the minimal clinically important difference for NRS [37]), the intensity was adjusted, and additional series of stimuli were performed until the pain intensity was reported at the same intensity as that induced by stimulation at L3. This technique was used to maintain the amplitude of reported pain at each site. To avoid habituation of reflexes, stimuli were provided at pseudo-random intervals between 5–30 s.

### Procedure

Participants performed maximal isometric voluntary contractions (MVC) of each muscle against manual resistance for 3 s in different positions for each muscle group: trunk extension

in prone lying (LM, TES), trunk flexion in supine lying (RA, OI, OE), and right/left trunk rotation in supine lying (OI, OE). Details of the procedure are reported elsewhere [38]. MVC EMG was used for EMG normalization and to calibrate the visual feedback of LM EMG to control the EMG amplitude in sitting trials (see below).

For each participant, the NWR was measured in side-lying and sitting. In side-lying, participants were positioned with pillows to support their head and between their knees to maintain a "neutral" spinal alignment. In sitting, participants sat on a bench without a backrest with their feet flat on the floor. To standardize the baseline EMG between participants and trials in sitting, participants performed gentle anterior pelvic tilt to activate LM at 10% of MVC with feedback of LM EMG on a computer monitor placed in front.

For both body positions, stimuli were applied to four sites: S1, L3, T12, and Right Rib. A series of six noxious stimuli were delivered at each site, for a total of 24 stimuli in each body position. To ensure that pain was constant across sites, pain was reported using the NRS after each stimulation and the intensity of stimulation was adjusted to maintain the targeted pain intensity, when necessary. The order of body positions and stimulation sites were randomized for each participant. Any symptoms of LBP at rest were reported using the NRS both before and after each block of stimuli in each body position.

## EMG analysis

EMG data were analysed using Matlab R2015b (The Mathworks, MA, USA). The artifact produced by electrical stimulation affected some EMG recordings but could generally be removed using a bandpass filter of 70–500 Hz (zero lag, $5^{th}$ order, Butterworth filter). Although this filtering removes some of the frequency band that includes EMG, and would reduce the signal amplitude, when all data are filtered in the same manner the relative differences between participants and tasks are preserved with respect to analysis undertaken on data filtered at the conventional 20–500 Hz ($2^{nd}$ order Butterworth) [26]. To avoid biasing the results all EMG was filtered in this manner. A similar technique (200–1000 Hz bandpass filtering) has been used to remove signal derivation and to successfully measure F-wave amplitude [39]. Three outcomes were measured from the NWR EMG responses: (i) late reflex response amplitude–RMS EMG amplitude between 80 to 200 ms, (ii) earliest onset latency of the reflex, and (iii) the frequency of occurrence of the early response (<80 ms). The early reflex component response amplitude was not measured since many participants with CLBP required high current intensity that elicited large stimulation artifact that makes the analysis invalid. Analysis windows were selected based on the outcomes of the previous study of pain-free individuals [26].

For calculation of the late reflex amplitude, the RMS EMG prior to the stimulus (110 to 10 ms prior to electrical stimulation) was subtracted from NWR late RMS EMG amplitude, and then presented as a proportion of the MVC EMG for each muscle. Subtracting the pre-stimulus EMG was essential to control for variation in baseline EMG prior to noxious stimulation. The MVC normalization technique is recommended by the Consensus for Experimental Design in Electromyography (CEDE) project to allow between-group and between-muscle comparisons [40]. The latency of the NWR was calculated as the time between the beginning of the electrical stimulation and the onset of the NWR. To determine whether a response was present, and its onset, each EMG signal was displayed using a graphical user interface (Matlab). A response was identified automatically when the EMG amplitude exceeded the mean of the pre-stimulus EMG activity (100-ms window) by 1 standard deviation for 50 ms, and this was confirmed visually [41]. We calculated the frequency of occurrence as a ratio between the number of identifiable early responses divided by the total number of stimulations.

## Statistical analysis

Data were analyzed using Statistical Program for the Social Sciences (SPSS 25, IBM Corp, Armonk, NY). Sample characteristics and NWR threshold were compared between groups (CTL vs. CLBP) using Mann Whitney U test for independent samples or Chi-square/Fisher's Exact test, because of the small sample size.

To evaluate differences in *motor strategies* and *receptive field* between groups we compared each muscle between groups for each site of noxious stimulation. The models included Groups (CTL vs. CLBP; between group) and stimulation Sites (S1, L3, T12, right rib; repeated measure) using Generalized Estimating Equations (GEE) separately for each outcome (EMG amplitude late reflex response, frequency of early response) and for each muscle. In addition, pain reported in response to the stimuli at each stimulation site was compared between groups. A model-based covariance matrix (because of the small number of factors used in the model [42]) and an exchangeable working correlation matrix were used. The validity of the working correlation matrix selected was confirmed using the Quasi-likelihood under Independence model information Criterion. The normality of the distribution of the residuals was verified using a QQ plot. If the residuals were not normally distributed a transformation was applied to transform negative values (e.g., reduction in EMG following electrical stimulation) and overcome the positive skew and the leptokurtotic nature of the non-transformed distribution. A posthoc Wald chi-square test with Bonferroni correction for multiple comparisons was used (adjusted p-values are reported) for pairwise posthoc comparisons. We interpreted *receptive field* and *motor strategy* differences only in presence of significant Group x Site interactions. Differences in *motor strategy* were considered in presence of significant posthoc comparisons between *groups* (i.e., between-group comparisons) for each site and muscle. Differences in *receptive field* were considered in presence of significant posthoc comparisons in reflex amplitude/occurrence between *sites* (i.e., within-group comparison), for each muscle and group.

Considering that participants with CLBP had large between-subject variability for the NWR threshold in lying, and this could be delineated into two sub-populations that responded uniquely, we performed additional analyses. One group responded similarly to CTL group ("Low-threshold group", n = 7), whereas another group presented with a substantially higher NWR threshold ("High-threshold group", n = 5). To classify participants in each of these subgroups, we used the upper limit of 95% confidence interval of the NWR threshold of the CTL sample (13.6 mA) as the cut-off value. These additional analyses were undertaken because most of the results were explained by differences between the CTL and "High-threshold" groups (especially for abdominal response to noxious stimulation). Similar statistical analyses were computed to determine whether differences were present between subgroups with the Group factor replaced by Subgroup (CTL, High-, Low-threshold). The current low back pain intensity reported before and after each block of stimuli in each body position was compared between the Low and High-threshold subgroups using Mann-Whitney U non-parametric statistics.

Significance was set at p<0.05. Data are presented as mean (standard error of the mean; SEM = standard deviation (SD)/$\sqrt{n}$)) throughout the text and figures without transformation to aid visualization and interpretation of the data.

## Results

The anonymized dataset is available from the Scholars Dataverse repository (https://doi.org/10.5683/SP3/OVTYIN). Table 1 reports the characteristics of participants in groups (CLBP vs. CTL) and CLBP subgroups (Low- and High-threshold). In two participants no NWR was

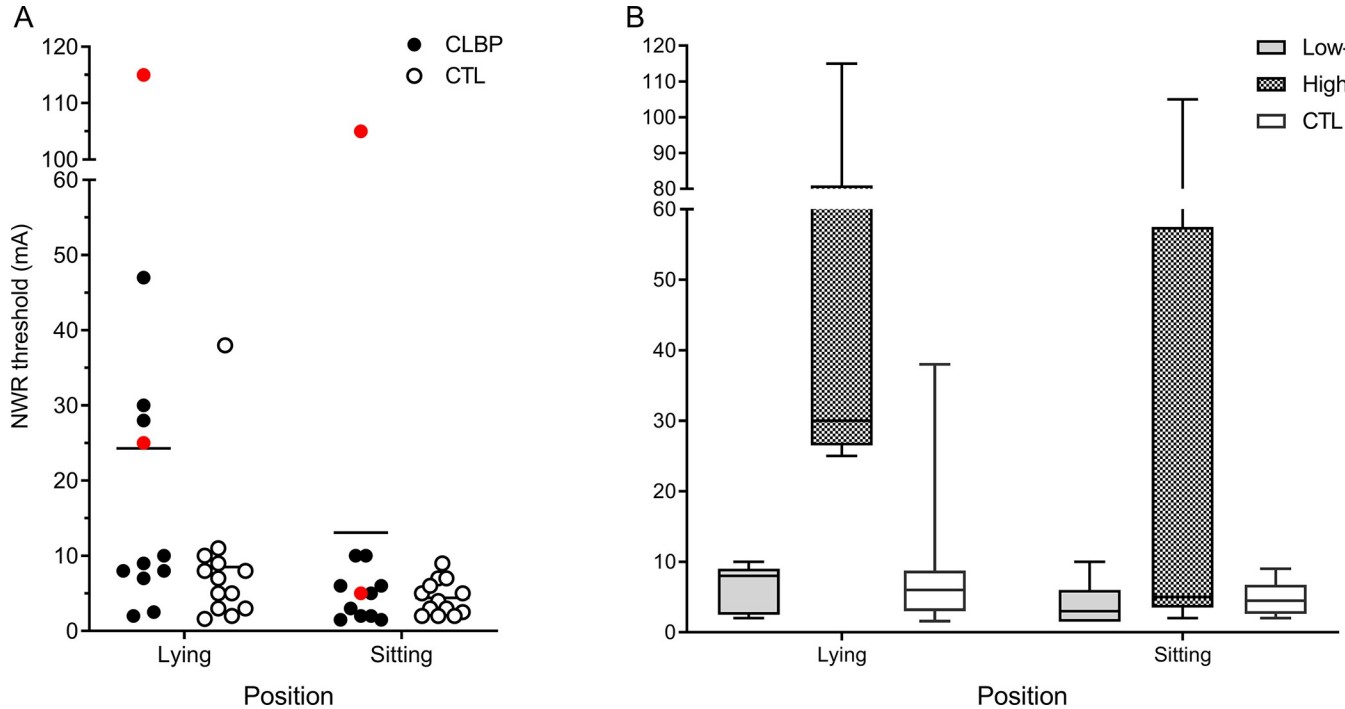

**Fig 1.** NWR threshold for (A) individual participants in CLBP and CTL groups, and (B) box and whisker plot of NWR threshold in CTL, and Low- and High-threshold subgroups. Red circles indicate participants in whom the NWR threshold was tested using TES because no response was observed in LM. Note the large variability in CLBP and the clear separation into Low- (<10 mA) and High- (>20 mA). NWR: Nociceptive withdrawal reflex; CLBP: Chronic low back pain; CTL: Pain-free controls. Box and whisker plots depict 2.5 to 97.5 percentiles, and the line represents the median.

identified for LM with stimulation at 100 mA and the parameters of stimulation were determined using the NWR threshold of TES (Fig 1A, red circles). LBP did not increase during the experiments (p>0.10).

Table 2 presents the pain evoked by noxious stimulation for each site in both positions for groups and subgroups; there was no main effect or interaction (all: p>0.09). Although the amplitude of the early response could not be assessed because of the presence of EMG artifacts, it was possible to determine if an early response was present. As the distributions of the frequency of the occurrence of motor responses and the variation in amplitude of the early responses varied in a similar manner between sites and muscles [26], the

**Table 2. Reported pain (NRS–out of 10) at different locations in lying and sitting by groups and subgroups (mean (SD)).**

|         | Location | CTL       | CLBP      | Low-threshold | High-threshold |
|---------|----------|-----------|-----------|---------------|----------------|
| Lying   | S1       | 6.2 (3.2) | 7.1 (2.2) | 6.9 (0.7)     | 7.3 (2.8)      |
|         | L3       | 5.8 (3.2) | 6.9 (2.2) | 6.8 (1.7)     | 6.9 (3.1)      |
|         | T12      | 6.0 (3.0) | 6.4 (2.3) | 6.3 (2.2))    | 6.6 (2.8)      |
|         | Rib      | 6.0 (3.2) | 6.9 (2.1) | 6.7 (1.7)     | 7.2 (2.8)      |
| Sitting | S1       | 5.7 (3.0) | 5.5 (2.7) | 5.1 (2.5)     | 6.1 (3.2)      |
|         | L3       | 5.5 (3.0) | 5.6 (2.6) | 5.4 (2.2)     | 5.8 (3.3)      |
|         | T12      | 5.5 (3.0) | 5.4 (2.6) | 5.1 (2.2)     | 5.8 (3.3)      |
|         | Rib      | 5.4 (2.9) | 5.4 (2.9) | 5.0 (2.6)     | 6.0 (3.5)      |

SD: Standard deviation; S1: First sacral vertebra; L3: Third lumbar vertebra; T12: Twelfth thoracic vertebrae.

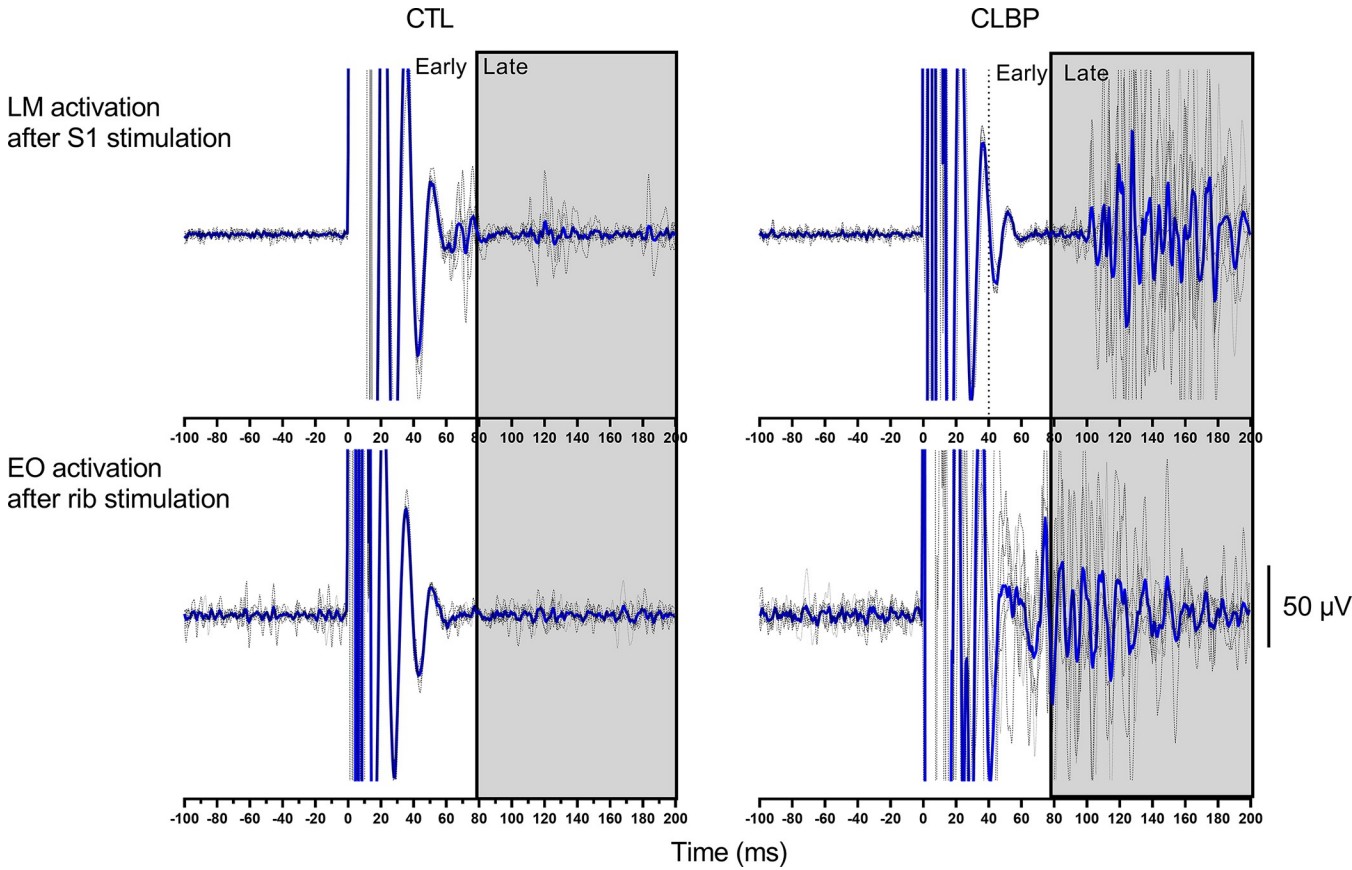

**Fig 2.** Examples of EMG activation of LM after S1 stimulation (upper panels) and EO after rib stimulation (lower panels) in controls (left panels) and participants with CLBP (right panels). The blue traces represent the EMG trace of all stimulation trials averaged. The grey traces represent single trials. The grey boxes represent the late windows (80–200 ms). Large artifact influences EMG signal although it did not impede the identification of early reflex occurrences. LM: Lumbar multifidus; OE: Obliquus externus abdominis; S1: First spinous process of the sacrum.

frequency of occurrence of the early responses were also analyzed. Fig 2 shows EMG signals from one pain-free control and one participant with CLBP. Note that despite the large artifact that may influence the early EMG window, it is possible to identify the occurrence of responses.

## NWR threshold–excitability of the L3 NWR

Despite the large apparent difference of the average NWR thresholds between groups in lying (CLBP: 24.3 ± 9.1 mA; CTL: 8.3 ± 2.8 mA), the difference was not significant (p = 0.06). Fig 1A shows that this was explained by the large variability of the NWR threshold in the CLBP group, particularly in lying, and the two distinct subgroups with different thresholds (Fig 1B; see methods; High- [41.0 ± 16.9 mA] and Low-threshold [6.6 ± 1.2 mA]). In sitting, feedback of LM EMG was provided to the participant to standardize the baseline activation level. This may have limited the variation in the facilitation of the motoneuron pool and its impact on threshold. In CLBP, the latency of the trunk muscle responses elicited by noxious stimuli varied between 77.5 ms (OE following rib stimulation in lying) to 154.3 ms (RA after S1 stimulation in lying). For controls, the latency ranged from 79.6 ms (OE after rib stimulation in lying) to 165.0 ms (RA after S1 stimulation in lying).

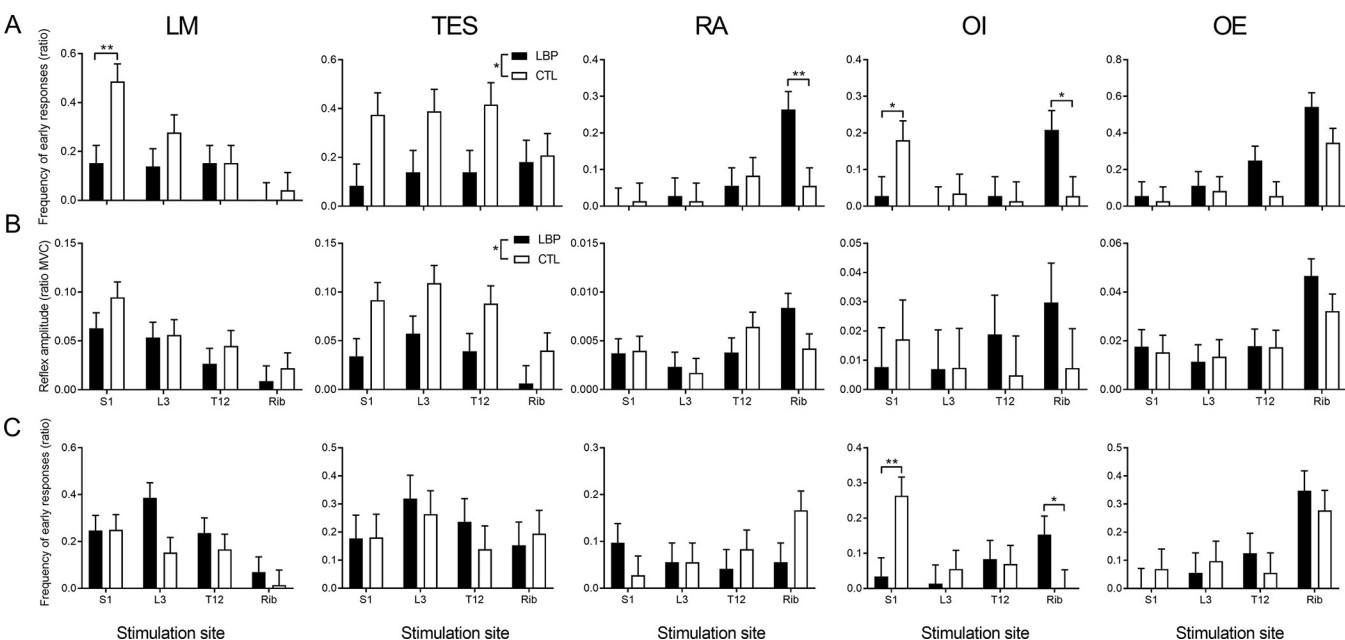

**Fig 3.** Trunk muscles responses after noxious stimuli at the four trunk sites in lying (A, B) and sitting (C) for both groups. (A) Frequency of occurrence of early responses and (B) amplitude of late responses in lying, and (C) occurrence of early responses in sitting for LM, TES, RA, OI and OE. Group x Site interactions were present in (A) for LM, RA and OI and (C) for OI. Group main effects were present in (A) and (B) for TES. An early response is a response occurring < 80 ms after the stimulation. CTL: Controls; LBP: Low back pain; LM: Lumbar multifidus; TES: Thoracic erector spinae; RA: Rectus abdominis; OI: Obliquus internus; OE: Obliquus externus; S1/L3/T12: Spinous processes of S1, L3 and T12. Ratio: Number of identifiable responses divided by the total number of trials. *p<0.05; **p<0.01; ***p<0.001.

## Between-group comparison of motor strategies and receptive field induced by noxious stimulation

For the presentation of the results, between-group pairwise comparisons for significant Group × Site interactions are reported in Fig 3, and GEE statistics and between-site pairwise comparisons are reported in Table 3. Complete subgroup analyses (Subgroup × Site models)

**Table 3. Between-location, within-group pairwise comparisons for significant Group × Location interaction.**

|  | Position | Muscle | Wald $\chi^2$; p | Site |  | CLBP |  | CTL |  |
|---|---|---|---|---|---|---|---|---|---|
| Occurrence of early response | Lying | LM | 11.0; 0.01 | S1 | L3 | - | - | 0.04 | ↑ |
|  |  |  |  |  | T12 | - | - | <0.001 | ↑ |
|  |  |  |  |  | Rib | - | - | <0.001 | ↑ |
|  |  |  |  | L3 | Rib | - | - | 0.01 | ↑ |
|  |  | RA | 8.9; 0.003 | Rib | S1 | 0.001 | ↑ | - | - |
|  |  |  |  |  | L3 | 0.003 | ↑ | - | - |
|  |  |  |  |  | T12 | 0.01 | ↑ | - | - |
|  |  | OI | 10.2; 0.02 | Rib | L3 | 0.03 | ↑ | - | - |
|  | Sitting | OI | 14.5; 0.002 | S1 | L3 | - | - | 0.02 | ↑ |
|  |  |  |  |  | T12 | - | - | 0.04 | ↑ |
|  |  |  |  |  | Rib | - | - | 0.002 | ↑ |

CLBP: Chronic low back pain; CTL: Control group; LM: Lumbar multifidus; OE/OI: Obliquus externus/internus; RA: Rectus Abdominus; S1; Sacrum; L3/T12; spinous process of L3/T12; ↑/↓: Muscle response amplitude/occurrence following stimulation of the site in the left column is larger/smaller than the after stimulation of the site in the right column.

are fully described in the Supporting information. We highlighted in the following sections differences in subgroup analyses that explained group differences.

## Motor strategies–lying

Group × Site interaction was significant for the frequency of occurrence of the early response of LM (Wald $\chi^2$ (3) = 11.0; p = 0.01), RA (Wald $\chi^2$ (3) = 8.03; p = 0.045) and OI (Wald $\chi^2$ (3) = 10.2; p = 0.02).

*Motor strategies* represent between-group differences in the occurrence/amplitude of muscle responses after stimulation of a given site by posthoc analyses of significant interactions. For stimulation of the rib, early responses of abdominal muscles (RA [p = 0.003] and OI [p = 0.002]) were more frequent for CLBP than CTL (Fig 3A). S1 stimulation was characterized by less frequent LM (p = 0.001) and OI (p = 0.04) responses in CLBP than in CTL.

Early responses of abdominal muscles (RA, OE and OI) were more frequent (S1A Fig) for High- than Low-threshold CLBP (occurrence: OI: p<0.001; OE: p<0.001) and CTL (occurrence: RA: p<0.001; OI: p<0.001; OE: p<0.001) after rib stimulation.

A main effect of Group represents an effect across all sites. The occurrence of TES early response (Main effect: Group—Wald $\chi^2$(1) = 4.4; p = 0.04—Fig 3A) and the amplitude of TES late response (Main effect: Group—Wald $\chi^2$(1) = 5.9; p = 0.02—Fig 3B) were smaller in CLBP than CTL.

## Receptive field–lying

*Receptive fields* were considered by within-group comparisons of the occurrence/amplitude of responses of a given muscle across stimulation sites independently for each group. For example, Fig 3 presents the *receptive field* of each trunk muscle separately and Table 3 reports differences between sites for each muscle and group, separately and detailed p-values for all within-group comparisons. Occurrences of early responses of LM presented different patterns between CLBP and CTL suggesting different muscle *receptive fields* (see Fig 3A; Table 3). For example, more frequent LM responses after S1 stimulation in lying were observed (i.e., the stimulus from which the LM can most effectively withdraw the body) than after stimulation of any other sites in CTL (all p<0.04), whereas no between-site differences were observed in CLBP (Fig 3A; Table 3). The occurrence of the early RA responses in lying was higher following rib stimulation than after stimulation of any other sites (all p<0.01) whereas no between-site difference was present in CTL (Table 3). The more frequent occurrence of early RA responses following rib stimulation than any other site was present only in the high-threshold CLBP subgroup (S1A Fig; Table A in S1 Text).

## Motor strategies–sitting

There was a significant Group × Site interaction for the frequency of occurrence of early responses of OI (Wald $\chi^2$ (3) = 14.5; p = 0.002).

Early OI response was evoked more frequently by the stimulation of S1 (p = 0.002) and less frequently by the stimulation of the rib (p = 0.04) in CTL than CLBP (Fig 3C). The more frequent early OI responses following rib stimulation were explained by the high-threshold CLBP subgroup (S2 Fig; S1 Table in S1 Text). Indeed, more frequent OI early responses were elicited by rib stimulation in high-threshold compared to low-threshold (p = 0.008) and CTL (p = 0.001).

### Receptive fields—sitting

The organization of the *receptive field* in sitting differed between groups at some stimulation sites (see Table 3 for detailed statistics). For example, the frequency of occurrence of early OI responses was larger following the stimulation of S1 compared to the stimulation of other sites in controls (all p<0.04) but not in participants with CLBP.

## Discussion

This exploratory study aimed to determine whether excitability and organization of the trunk NWR differed between individuals with and without CLBP. Contrary to our first hypothesis, the NWR threshold of the lumbar region was not lower. Instead, it was either similar to, or higher than, that of controls as described by the presence of two subgroups of participants with CLBP. Consistent with our second hypothesis, we observed differences in the organization of the trunk NWR between groups. Overall, participants with CLBP presented with less frequent NWR of LM with S1 stimulation, despite very high stimulus intensity being used in some participants with CLBP. Also, early abdominal muscle responses were more frequent and this was driven by participants with CLBP and higher NWR threshold. These results are not consistent with a simple increase in central nervous system excitability in CLBP, but rather a complex and different organization of the trunk NWR in presence of CLBP, including hyposensitivity in some individuals along with changes in *motor strategies* and *receptive fields*. Due to the small sample size, non-significant results can be the results of insufficient power to detect differences and should be interpreted cautiously.

### Limitations

The results of this exploratory study must be considered with respect to methodological issues. First, the CLBP group was young and reported low disability, which might limit the generalizability of the results. Second, large responses of abdominal muscles to rib stimulation in the high-threshold subgroup might be evoked by direct intercostal nerve stimulation. Direct intercostal nerve stimulation evokes OE M-waves with short latency [21,43]. We consider this unlikely to explain our observations as no similar large abdominal muscle responses were identified in the control participant who required high intensity stimulation to evoke the NWR. Third, we did not measure the early response amplitude due to contamination of the early response by electrical stimulation artifact for many participants with CLBP. We reduced the amount of lost data using a higher than usual high-pass filter to the EMG. This will have removed some energy from the signal, which requires consideration, but was consistent for the whole sample. To analyse the early component of the reflex, we evaluated the frequency of occurrence of the early response [26] considering the similarity between the distributions of the frequency of occurrence motor and response amplitude for a given combination of muscle and stimulation site (e.g. a muscle with lower occurrence had often a small EMG amplitude). Fourth, the electrical stimulation was done using a handheld probe that may have slightly influenced the afferent recruited during the stimulation.

### Sensitization in CLBP

Individuals with CLBP were not more sensitive to the noxious stimulus over the lumbar area. Pain response to stimulation did not differ, even in the subgroup of participants with higher NWR threshold and despite higher stimulation intensity. This contrasts with studies that have identified lower NWR thresholds when stimuli are applied to the foot in CLBP, suggesting central sensitization [27–30]. Previous studies of acute LBP [44] and other chronic pain states [45]

have been unable to induce NWR in response to noxious stimulation of the foot in 30–32% of participants, whereas NWR could be evoked in all painfree participants [46]. This concurs with the subgroup of participants with a high threshold and may explain the discrepancies with our results. The NWR threshold was high at >25 mA in 5/12 individuals with CLBP, including 2 participants without LM response (TES was used instead to identify the NWR threshold). In humans, no study has tested NWR within the clinical pain area. A study that reported intracutaneous and intramuscular electrical stimulation to the lower back to assess pain threshold showed no between-group difference but observed large variability within the CLBP group [47], similar to our study.

Lower back pressure and heat pain threshold also present conflicting results in CLBP [31]. Although some studies report lower pressure pain thresholds [48–51], others do not [31], and changes in heat pain threshold have been rarely reported [52]. Although sensitization in the clinical pain area may have been expected, this is not uniformly supported by the present or previous data [31,53,54]. Heterogeneity of CLBP is expected. Recent studies described different clusters of individuals with CLBP based on different sensory profiles; some with hypersensitivity, whereas others did not differ from painfree individuals [53,55]. This latter cluster concurs with 7 participants out of 12 (low-threshold subgroup) who presented similar NWR thresholds compared to CTL. No study has reported hyposensitivity for pressure and temperature thresholds. This may imply that NWR threshold properties differ from pressure and temperature pain thresholds and assess different nociception/pain features. The absence of sensitization in our sample might be explained by the characteristics of CLBP participants. Our participants were young, reported low disability (mean ODI—18%) and no CLBP participants scored >40 on the Central Sensitization Inventory [56], i.e. the cut-off for clinical symptoms of central sensitization [57]. Individuals with prominent clinical symptoms of central sensitization (i.e. nociplastic pain presentation [58]) might respond differently to noxious stimuli.

As the NWR of the foot sole was not assessed, we cannot compare results with existing literature. However, considering that the Rib is remote from the area of clinical pain (low back), over-activation of abdominal muscles may suggest sensitization in CLBP and our additional analyses point more specifically toward the high-threshold subgroup. Considering that NWR threshold was not individually assessed at each stimulation site, it is unclear whether the higher abdominal muscle responses were due to the higher intensity of stimulation used in CLBP than CTL or a lower rib NWR threshold in CLBP. It is noteworthy that the pain elicited by the electrical stimulus was not reported to differ between CLBP and CTL despite the use of larger intensity of stimulation in CLBP. This could be clarified in future studies with NWR threshold identified at all sites to enable greater confidence in between-site and between-group comparisons.

This is the first report of lower reactivity to local noxious stimuli in CLBP. In addition, we observed fewer occurrences of erector spinae (LM and TES) responses even though very high intensities of noxious stimulation were used in some participants with CLBP. A plausible explanation for the high NWR threshold is that the elicited low back movement might be provocative of pain in some participants. Although speculative, the central nervous system might suppress/reduce the NWR to avoid pain expected to be induced by movement. That might be expected if the movement of the spine to withdraw from the stimulus was more threatening than the noxious stimulus. Suppression (higher threshold or less frequent responses of erector spinae muscles) of the NWR might be a protective and adaptive mechanism.

### Re-organization of trunk muscle NWR receptive fields and motor strategies

Organization of the trunk muscle NWR *receptive fields* and *motor strategies* differed in CLBP. Two broad patterns of responses adaptation were observed in lying and somewhat in sitting:

(i) more frequent abdominal muscle responses following 8[th] rib stimulation (explained exclusively by the participants with high threshold), and (ii) less frequent and smaller responses of erector spinae muscles suggesting modified *motor strategies* in CLBP. These findings partially concur with observations of two extremes of adaptation in trunk muscle activation in CLBP during other tasks. For instance, studies have observed increased oblique abdominal muscle activation [2,59] in CLBP. Further, delayed [3,4,60] of LM activation and increased TES activation [61] have been reported frequently. As mentioned previously, our results might suggest suppression of the NWR to limit potentially painful spine movement despite very high stimulation intensity. All abdominal muscles had larger amplitude in CLBP after stimulation of the 8[th] rib regardless of the biomechanical role of these muscles. Although some muscles would move the trunk away from the stimulus (e.g., withdrawal by OE in painfree participants [26] and increased in CLBP), other abdominal muscles that were minimally involved in withdrawal following 8[th] rib stimulation in CTL (OI, RA) were also more frequently activated. This might represent a strategy of trunk stiffening (co-contraction with antagonist muscles) rather than moving the spine away from the stimulus. Differences in *receptive fields* were also observed in CLBP. For example, rib stimulation elicited more frequent RA motor responses than any other sites of stimulation in CLBP whereas no difference between sites was present in CTL. Also, LM (in lying) and OI (in lying and sitting) responses were tuned to the site of stimulation only in CTL as already reported in our previous study [26]. Both LM and OI responses were more frequent following S1 stimulation than after stimulation of any other sites which emphasizes their roles in extension of the lower lumbar spine and control of the lumbar lordosis [62,63]. In CLBP, this site-dependent modulation was absent.

Most between-group differences were observed in the early reflex window. The short latency of the NWR of the abdominal [21] and paravertebral [26] muscles suggests a spinal origin, as there is insufficient time for transcortical relay [21]. The different trunk muscle patterns observed in CLBP may reflect a different organization of spinal networks controlling trunk muscles. The specificity of the *motor strategies* to the site of stimulation highlights that spinal networks contain sufficient complexity to organize this *motor strategy* [26], and the present data highlight that this organization differs in CLBP. Our results suggest that the structure of the response is changed in terms of the *motor strategies* and *receptive fields*. These complex responses to noxious stimuli depend on the site stimulated, and perhaps the perceived threat of the movement induced by the noxious stimulus. These results are the first to suggest a different organization in motor spinal networks in humans and add to the evidence of altered central nervous system motor pathways [6–10,64,65]. Also, it is suggested that M1 may contribute to the top-down control of nociception and pain processing (see references [66,67]), and it could in part explain differences observed in NWR in CLBP. Differences in the organization of spinal networks controlling trunk muscles are likely to contribute to the alteration in spine motor control in CLBP [3,61,68] in conjunction with changes in supraspinal areas [13].

## Conclusion

This study shows different organization of trunk muscle responses to noxious stimulation at multiple trunk sites in CLBP. More frequent abdominal muscle responses after rib simulation, and smaller and less frequent erector spinae muscles suggest a re-organization of spinal networks controlling trunk muscles in CLBP. In addition, a subgroup of participants with CLBP had a high NWR threshold to stimuli applied to the clinical pain area which suggest that interpretation of *sensitization* varies between individuals with CLBP and cannot be generalized to all body sites. Due to the small sample size, replication of our results with a larger sample size would be beneficial.

## Supporting information

**S1 Fig. Activation of trunk muscles following noxious stimulation of the four trunk locations in lying.** (A) Frequency of occurrence of early responses(ratio), and (B) amplitude of late responses for LM, TES, RA, OI and OE. LM: Lumbar multifidus; TES: Thoracic erector spinae; RA: Rectus abdominus; OI: Obliquus internus abdominis; OE: Obliquus externus abdominis; S1/L3/T12: Spinous processes of S1, L3 and T12. $*p<0.05$; $**p<0.01$; $***p<0.001$.
(TIF)

**S2 Fig. Activation of trunk muscles following noxious stimulation of the four trunk locations in sitting.** (A) Frequency of occurrence of early responses (ratio), and (B) amplitude of late responses for LM, TES, RA, OI and OE. LM: Lumbar multifidus; TES: Thoracic erector spinae; RA: Rectus abdominus; OI: Obliquus internus abdominis; OE: Obliquus externus abdominis; S1/L3/T12: Spinous processes of S1, L3 and T12. $*p<0.05$; $**p<0.01$; $***p<0.001$; X—Subroup x Site interaction was observed but without Subgroup differences detected by pairwise comparisons, and comparisons between locations (within-group) are reported in Table A in S1 Text.
(TIF)

**S1 Text. Detail of the results from the subgroup analysis including S1 Table.**
(DOCX)

## Author Contributions

**Conceptualization:** Hugo Massé-Alarie, Paul W. Hodges.

**Data curation:** Hugo Massé-Alarie, Genevieve V. Hamer.

**Formal analysis:** Hugo Massé-Alarie, Sauro E. Salomoni, Paul W. Hodges.

**Funding acquisition:** Paul W. Hodges.

**Investigation:** Hugo Massé-Alarie, Genevieve V. Hamer.

**Methodology:** Hugo Massé-Alarie, Genevieve V. Hamer, Sauro E. Salomoni, Paul W. Hodges.

**Project administration:** Hugo Massé-Alarie, Paul W. Hodges.

**Resources:** Paul W. Hodges.

**Software:** Sauro E. Salomoni, Paul W. Hodges.

**Supervision:** Paul W. Hodges.

**Visualization:** Hugo Massé-Alarie, Paul W. Hodges.

**Writing – original draft:** Hugo Massé-Alarie.

**Writing – review & editing:** Hugo Massé-Alarie, Genevieve V. Hamer, Sauro E. Salomoni, Paul W. Hodges.

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
