## [Decision Letter · Decision Letter 0]

26 Oct 2022

PONE-D-22-26918Nociceptive withdrawal reflexes of the trunk muscles in chronic low back painPLOS ONE

Dear Dr. Massé-Alarie,

Thank you for submitting your manuscript to PLOS ONE. After careful consideration, we feel that it has merit but does not fully meet PLOS ONE’s publication criteria as it currently stands. Therefore, we invite you to submit a revised version of the manuscript that addresses the points raised during the review process. As you will see below, several points were raised by the Reviewers during the review processes. Reviewer #1, in particular, points to specific methodological issues in the study protocol. For instance, there are questions about the sample size, the selection of stimulation and recording sites and the method to determine sensory thresholds. The Reviewer also has relevant suggestions for the presentation of the results (e.g., addition of EMG traces of reflex responses). Along the same line, Reviewer #2 has some suggestions to improve the discussion. Please make sure that all issues are adequately addressed in the revised version.

We look forward to receiving your revised manuscript.

Kind regards,

François Tremblay, PhD

Academic Editor

PLOS ONE

“This study was funded by a Program Grant from the National Health and Medical Research Council of Australia (NHMRC) of Australia (APP1091302)**. **HMA was supported by a Postdoctoral Fellowship from the Canadian Institutes for Health Research (358797) and is supported by a Research Scholar Awards from Fonds de recherche du Québec – Santé (HMA: 281961). P.H. is supported by a Fellowship (APP1194937) from the NHMRC. The funders had no role in study design, data collection and analysis, decision to publish, or preparation of the manuscript. None of the authors have potential conflicts of interest to be disclosed.”

“This study was funded by a Program Grant from the National Health and Medical Research Council of Australia (NHMRC) of Australia (APP1091302) awarded to PH. HMA was supported by a Postdoctoral Fellowship from the Canadian Institutes for Health Research (358797) and is supported by a Research Scholar Awards from Fonds de recherche du Québec – Santé (HMA: 281961). P.H. is supported by a Fellowship (APP1194937) from the NHMRC. The funders had no role in study design, data collection and analysis, decision to publish, or preparation of the manuscript. None of the authors have potential conflicts of interest to be disclosed.”

3. We noted in your submission details that a portion of your manuscript may have been presented or published elsewhere. “Yes, data form pain-free participants were published in European Journal of Neuroscience. This is not a dual publicaiton since the objectives of the two papers are different. In this paper, we compare data collected in participants with chronic low back pain to the painfree control group. This objective of the paper published in European Journal of Neuroscience was to determine if stimulation of different sites on the trunk induced different pattern of trunk muscle activation”. Please clarify whether this publication was peer-reviewed and formally published. If this work was previously peer-reviewed and published, in the cover letter please provide the reason that this work does not constitute dual publication and should be included in the current manuscript.

Reviewers' comments:

Reviewer's Responses to Questions

**Comments to the Author**

1. Is the manuscript technically sound, and do the data support the conclusions?

Reviewer #1: Partly

Reviewer #2: Yes

2. Has the statistical analysis been performed appropriately and rigorously? 

Reviewer #1: Yes

Reviewer #2: Yes

3. Have the authors made all data underlying the findings in their manuscript fully available?

Reviewer #1: Yes

Reviewer #2: Yes

4. Is the manuscript presented in an intelligible fashion and written in standard English?

Reviewer #1: Yes

Reviewer #2: Yes

5. Review Comments to the Author

Reviewer #1: In this manuscript the authors compare the NWR of the trunk between healthy individuals and individuals with low back pain. The study is highly interesting and addresses a large group of individuals, suffering from chronic pain, many often very poorly understood. Very few studies have investigated the NWR in areas close to chronic pain area, thus, making this study very novel. The results of the study suggest that individuals with CLBP can be divided into two group based on their NWR threshold.

In many aspects the study design and general methodology of the study seem sound. However, I have several concerns regarding the reflex methodology, and analysis hereof, which I am sure is not ideal and leaves substantial room for improvement. I am also surprised to see how small group sizes are used to compare individuals suffering from chronic pain.

General comments.

What is the rational for the number of individuals with chronic pain and controls included in the study? Were any power calculations or similar made? I notice that the group sizes are smaller than the authors previous work on the trunk NWR (Massé-Alarie et al, EJN 2019), however, that study was purely conducted in healthy volunteers. When conducting a study in individuals with chronic pain I would expect that a larger sample is needed compared to healthy controls, especially when doing the division into low/high threshold further limiting the sample size in this groups.

In this study the authors give electrical stimulation at several sites on the trunk to elicit NWR in the trunk. Why was the reflex threshold based purely on the stimuli in L3 and the reflex response from LM? How can the authors be sure that there no individual differences in reflex sensitivity across different stimulation sites. Is the combination of L3 and LM somehow representative of all the different stimulation sites and muscles investigated? For NWR studies of the foot, the stimulation intensity is always adjusted individually for each stimulation site, partly because of the variation in skin/subcutaneous anatomy (which I believe is probably also the case for skin over the trunk), but also due to the skin-electrode interface varies greatly, and this must be taken into account.

Please explain why the target muscle used for threshold estimation (LM) was not used in all subjects, but in some subjects, the TES was tested instead when reflexes in LM could not be elicited. Does this not mean that the subjects would have less reflex data, i.e. missing or highly reduced reflexes in LM? I fear that these subjects should have been excluded instead?

Please elaborate on how NWR thresholding was made, several studies have investigated the methodology of optimizing the NWR threshold determination (I suggest that the author see the work by Rhudy and France, see also specific comments below).

Please specify how many stimuli/reflexes have been omitted to stimulus artefacts.

The details about the ‘2-branch probe’ stimulation electrode must be clarified, possibly with a figure. How was the electrode fixed to the skin? Is this a similar electrode to that used by (Rukwied, Schmelz et al 2020) i.e. the positioning is handheld? If so, this means that the stimulation site alters with every stimulation at the ‘site’, resulting in slightly different afferents being stimulated and different impedances. Meaning that the stimulation intensity in relation to NWR threshold differs even more. Please clarify.

The manuscript should be expanded with examples of EMG traces from both CBLP and control. And possibly also with each subgroup of CBLP. I am curious to see / confirm the traces and whether there is background activity prior to electrical stimuli? I suggest the authors to illustrate the traces from the pre-stimulus period to end of the late response - possibly also including the electrical artefact, as a reader can differentiate that to actual EMG activity.

I believe that the supplementary material should be included as data in the main manuscript. Because this data is in my view better suited to answer the original hypothesis of the study (regarding NWR thresholds, and thus, the NWR responses as well, in my view). The subdivision of the LBP individuals is not directly related to the main hypothesis, but more the secondary hypothesis (see other comments).

The first part of the introduction and evidence that individuals with CBLP move differently should be clarified, so that it is clear how the trunk movement during CLBP is affected.

The manuscript is not very clear as how the stimulation intensity was adjusted over the course of the experiment. Initially an intensity of 2x NWR threshold was used, but this was then adjusted based on reported pain, this description is not sufficiently accurate, and does allow for reproducing the methods. How often was this adjusted, and to what degree? Was it done in all subjects?

Why was the reflex magnitude reported as normalized to MVC?

Please add more info about EMG electrode types, such as model name.

Why was the EMG amplification not adjusted to each individual? Individual anatomical difference, e.g. adipose layers etc may greatly affect the EMG signal and thus the need for amplification.

I find the results section(s) regarding the receptive fields to be unintuitive to understand. Fig 2 and 3 contain vast amounts of data but it is difficult to elude the details of the receptive fields based on these figures. If the authors wish to describe the receptive fields of different muscles (and the differences between CTL and CLBP), why not actually map the receptive fields based on the data (similar to Massé-Alarie, EJN, 2019) and then quantify the sizes and other characteristics of the RRF?

Please add statistic test to paratheses where p-values are described, both in the main text and figures/tables captions.

In first part of the discussion the conclusions regarding the two CBLP subgroups should be soften up, due to the low group sizes.

If compliant with Journal rules, I suggest changing the short title to “Different nociceptive withdrawal reflex in low back pain” I believe that adding ‘nociceptive’ clarifies the title.

There are several minor typo errors in the manuscript such as unneeded capital letters or missing capital letter (for example in figures), missing spaces between text and paratheses for reference etc. Additionally several sentences lack clarity, making it difficult to understand the meaning. Please proof-read the manuscript carefully.

Specific comments.

Abstract L 48 Please check sentence construction in ‘Two CLBP subgroups were identified by different in NWR threshold:’

Please be specific/consistent when using abbreviations. E.g. CLBP and LBP, I believe they are used too interchangeably to refer to pain sufferers. Would it better to simply use one?

L87-92 I do not believe that this secondary hypothesis is very well underlined, nor do I believe it is directly related to the subdivision of CBLP subjects (see later comments).

L133-137 what was the order of the analog EMG filter (BP 5Hz - 1kHz)? I assume it is ringing from these filters, which causes problems with the stimulation artefacts? I wonder if this could have been solved by adjusted the filter order, thus reducing the ringing.

P7 L147. I fear that NWR thresholding is not very systematic. The authors states ‘increase until a response was evoked’ what was the criteria for evoked reflex? I refer the authors but the works by Rhudy and France (2007/2009) etc on how to determine the threshold of the NWR. Additionally, in what steps were the intensity increased. There were no ‘staircase’ procedure or similar? (see also general comments above)

Fig 2B, please ensure y-label starts with capital letter.

L 143-4 I do not see the relevance or meaning of the sentence ‘Stimulation with 0.2-ms pulse width induces pain of 3-7/10 with stimuli of ~150mA to the quadriceps muscles (31).’ The electrical current is not easily translatable across different body regions, and different electrode configurations etc. I suggest clarifying or deleting.

P6 L 144. The phrase “This was perceived by the participant as a single noxious stimulus.” should have the word noxious removed.

L203-5 Why was this criterion for reflex detection used? What is the rationale for using this? Again I refer the authors to comprehensive work by Rhudy and France.

L208 The division of the LBP subject was made based on the data CTL and the 95 % CI. What is the rationale behind this, and how does this fit with the initial hypothesis? What other analysis were made to investigate this division?

L217-219 While I appreciate the authors interest in showing the level of variation in LBP. I urge the author to be careful with the conclusion based on the two subgroups of the LBP group because of the very small sample sizes. I am in doubt of whether this rather strong division is a general phenomenon in CLBP.

L253 Why is Site and Position written in capitals?

L 258-9 please clarify the sentence ‘…. could not be assessed, but the presence or not of an early response

was recorded.’

L372, delete ’ 6-fold’ this is not accurate for all cases.

L379-380 I strongly agree. Also why the conclusion regarding the subgroup must be soften up.

L391-392 this sentence is very hard to understand. In what way is the amplitude results the same as the number of occurrences?

L401 ‘This concurs with the response of the High-threshold group.’ I am not sure I agree with the authors here. In many of the studies being reference in sentences prior, common is that those subjects, where no reflexes could be elicited were often excluded. However, in the current study, subjects where reflexes were difficult to elicit, stimuli were merely given to another site to evoke reflexes from there. As mentioned above I am very doubtful of whether these subjects should have been included in the analysis.

L403 ‘In humans, no study has tested NWR within the clinical pain area. ’ this sentence is not correct. The authors reference several NWR studies made in chronic pain patients.

L422-423 the authors suggest that the high-thr have sensitization, creating this ‘overaction’. However, in case of CS that would mean lower NWR threshold (rather than higher threshold). Instead I believe this overaction of the abdominal muscles are due to the stimulation intensity being insufficiently calibrated to each stimulation site and the high-thr group using (too) high stimulation intensity.

The ‘Sensitization in CLBP’ section of the discussion is not very well structured and lacks focus, making it difficult to follow. Perhaps considering shortening, as I am not sure all content is relevant for the discussion of the current results.

Fig 1. Please add details on statistic. Was the data in 1A normally distributed?

Fig 2B + 3 B. Occurrence with large capital letter, but perhaps the word ‘Frequency’ is more appropriate? Please add units. If this is occurrence how can the number be less than 1 (this has not been explained in neither methods or captions). I suggest to report this in percentages. Add the definition of early reponse in the figure caption.

Generally Fig 2 and 3 are intended to allow comparison across stimulation sites and muscles. However, the y-scaling is not identical for any plots, greatly limiting the comparison across parameters. I strongly suggest the authors to re-consider this, especially as the main text tries to compare across sites and muscles.

Please add label for the x-axis for Fig 1-4, such as stimulation site. I recommend only doing so for the bottom row in each figure.

Reviewer #2: This exploratory study set out to determine whether the organisation and excitability of the trunk nociceptive withdrawal reflexes (NWR) are modified in chronic low back pain (CLBP). The authors hypothesised that individuals with CLBP would have modified

NWR pattern and lower NWR thresholds. Electrical stimulation was delivered over the spine at different vertebral levels and over the 8th rib in order to elicit NWR in 12 participants with CLBP and data compared with that collected previously from healthy participants, by the same authors, using an identical protocol. Surface electromyographic (EMG) activity was recorded from 5 trunk muscles and occurrence and sizes of responses were determined. The results showed that 2 CLBP subgroups were identified by differences in NWR thresholds, high and low. The responses in abdominal muscles were larger in those with high thresholds. Occurrence of response in multifidus were lower in CLBP groups than in controls. The authors conclude that the results suggest modified networks in spinal networks control trunk muscles could explain differences in motor control in those with CLBP.

The study is well written and experiments seem diligently performed. The results are novel and suggest differences in organisation of motor spinal networks in CLPB, which add to the evidence of changes in supraspinal areas. Caution is advised though as the n numbers were small in both the previously published control data and the current CLBP group.

I think it would be good to include some brief discussion of the idea that any changes in NWR thresholds and amplitudes might be related to the changes known to exist in supraspinal regions (e.g. motor cortex) in CLBP. In healthy participants, NWR induced by higher intensity stimulation appears to activate top-down analgesic mechanisms, which can be reduced with tDCS of the motor cortex (Hughes et al., 2019). NWR induced in those with ongoing CLBP are likely influenced by the co-existing changes within brain regions. There is only brief mention of the interplay between spinal and supraspinal circuits (lines 462-465) essentially stating that changes in spinal networks contribute to alterations in spine control along with changes in supraspinal areas.

Minor comments:

Line 65. I wonder if the sentence “Assessment of spinal motor networks that control back muscles is challenging” should be altered to say “trunk” instead of “back”, particularly since the preceding sentences in this paragraph related to the trunk, rather Line 72 – Change “controlled at the spinal cord” to “controlled at the level of the spinal cord”.

Line 73 – Change “and influenced by supraspinal control” to “and is influenced by supraspinal control”

Lines 76-77 – “in a manner consistent with its biomechanical function”. Does reference 21 apply to this too? If so, maybe it can be added at the end of this sentence?

Lines 204-205. “when the EMG amplitude exceeded the mean of the pre-stimulus EMG activity (100-ms window) by 1 standard deviation for 50 ms, and this was confirmed visually (35).” Although this is referenced, this seems an unusually low level to exceed in order to state that a response had occurred. Is this a mistake and it should be exceeding 2SDs?

Lines 428-431. This is quite speculative, is there any evidence that NWR can be intentionally suppressed?

Reference:

Hughes, S., Grimsey, S., Strutton, P.H. (2018). Primary motor cortex transcranial direct current stimulation modulates temporal summation of the nociceptive withdrawal reflex in healthy subjects. Pain Medicine. 20(6):1156-1165). doi: 10.1093/pm/pny200.

6. PLOS authors have the option to publish the peer review history of their article (what does this mean?). If published, this will include your full peer review and any attached files.

Reviewer #1: No

Reviewer #2: No

---

## [Author Response · Author response to Decision Letter 0]

9 Dec 2022

PONE-D-22-26918

Nociceptive withdrawal reflexes of the trunk muscles in chronic low back pain

Response to Reviewer #1:

In this manuscript the authors compare the NWR of the trunk between healthy individuals and individuals with low back pain. The study is highly interesting and addresses a large group of individuals, suffering from chronic pain, many often very poorly understood. Very few studies have investigated the NWR in areas close to chronic pain area, thus, making this study very novel. The results of the study suggest that individuals with CLBP can be divided into two group based on their NWR threshold.

In many aspects the study design and general methodology of the study seem sound. However, I have several concerns regarding the reflex methodology, and analysis hereof, which I am sure is not ideal and leaves substantial room for improvement. I am also surprised to see how small group sizes are used to compare individuals suffering from chronic pain.

RESPONSE: We thank Reviewer 1 for their comments on our work and we addressed their concerns in the following sections in red font.

General comments.

1. What is the rational for the number of individuals with chronic pain and controls included in the study? Were any power calculations or similar made? I notice that the group sizes are smaller than the authors previous work on the trunk NWR (Massé-Alarie et al, EJN 2019), however, that study was purely conducted in healthy volunteers. When conducting a study in individuals with chronic pain I would expect that a larger sample is needed compared to healthy controls, especially when doing the division into low/high threshold further limiting the sample size in this groups.

RESPONSE: Reviewer 1 is correct that the size of the sample size was based on feasibility of the recruitment. Considering that the participants with CLBP received noxious stimuli eliciting high intensity of pain, it was difficult to recruit them for the study. As a consequence, our study is limited to the detection of larger effects and we may have failed to detect more subtle differences between groups and it is possible that features that did not differ between groups might be found to differ with larger samples. We added this information in the Methods and acknowledged this limitation in the Discussion.

Methods:

p. 4-5; ln 107-109: “Considering the difficulty to recruit participants with CLBP in the study (noxious stimuli elicited high intensity of pain), the sample size was based on feasibility.”

Discussion

p.19; Ln 398-399: “Due to the small sample size, non-significant results can be the results of insufficient power to detect differences and should be interpreted cautiously.”

2. In this study the authors give electrical stimulation at several sites on the trunk to elicit NWR in the trunk. Why was the reflex threshold based purely on the stimuli in L3 and the reflex response from LM? How can the authors be sure that there no individual differences in reflex sensitivity across different stimulation sites. Is the combination of L3 and LM somehow representative of all the different stimulation sites and muscles investigated? For NWR studies of the foot, the stimulation intensity is always adjusted individually for each stimulation site, partly because of the variation in skin/subcutaneous anatomy (which I believe is probably also the case for skin over the trunk), but also due to the skin-electrode interface varies greatly, and this must be taken into account.

RESPONSE: We acknowledge that estimating a NWR threshold for each site would have been optimal for our study. We decided to use the same thresholds (1 in lying, 1 in sitting) for all the sites tested for several different reasons. First, determining the threshold was, by far, the longest part of the experiment. Considering that different participants had different thresholds (e.g., from 2 mA to 115 mA in lying), we started at very low intensity and increased the intensity by small increments. Second, we wanted to stimulate multiple sites (4 sites: 3 at the back; 1 at the rib) and recorded multiple trunk muscles (5 muscles: EO, IO, RA, TES, LES) around the trunk. Third, we searched for NWR threshold in two different positions (lying, sitting) for a total of 40 possible thresholds. Since it was not feasible to find a NWR threshold for each site in each position, we decided to individualise the intensity of stimulation using the pain reported at L3 and to match this pain intensity at each site. We think that matching pain intensity may better represent the threat of the noxious stimuli by the central nervous system and allows comparisons between groups. The pain reported by participants are presented in Table 2 and was not different between groups and subgroups.

We added some detail in Methods:

Methods

p. 7; Ln 163-171: “NWR threshold was only tested at L3 using LM motor responses. Considering the high number of painful stimuli and the time taken to find NWR threshold, it was not feasible to identify thresholds individually at each site and in both positions. To enable comparisons between sites, reported pain at L3 spinous process at 2 times NWR threshold was matched between sites by adjustment of stimulation intensity. At sites other than L3, participants rated the pain intensity following the first stimulus. If pain intensity differed from the target by 2 or more out of 10 (the minimal clinically important difference for NRS (35)), the intensity was adjusted, and additional series of stimuli were performed until the pain intensity was reported at the same intensity as that induced by stimulation at L3. This technique was used to maintain the amplitude of reported pain at each site.”

We selected the L3 and LM combination for several reasons. First, LM is an extensor of the lumbar spine and the principal muscle that has the potential to withdraw the lower back away from the L3 stimulation (2). Second, there is large body of evidence demonstrating that the LM is often altered in chronic low back pain (CLBP) (e.g.(3, 4)). Although the combination was selected for these reasons, we cannot be certain that NWR thresholds of other sites would be the same as for the L3/LM combination. However, considering that there was no difference in pain intensity across sites and groups, and there were two broad opposite patterns of results (reduction in back muscles; augmentation in abdominal muscles), we do not believe that the methods used explain our results. We acknowledge that we cannot ascertain whether results would be the same if we individualised NWR threshold at each site.

We added some detail in Discussion to highlight this issue:

Discussion

p. 21, Ln 449-455: “It is important to note we did not assess Rib NWR threshold individually. As such it is unclear whether the higher abdominal muscle responses were due to the higher intensity of stimulation used in CLBP than CTL or a lower rib NWR threshold in CLBP. It is noteworthy that the pain elicited by the electrical stimulus was not reported to differ between CLBP and CTL despite the use of larger intensity of stimulation in CLBP. This could be clarified in future studies with NWR threshold identified at all sites to enable greater confidence in between-site and between-group comparisons.”

3. Please explain why the target muscle used for threshold estimation (LM) was not used in all subjects, but in some subjects, the TES was tested instead when reflexes in LM could not be elicited. Does this not mean that the subjects would have less reflex data, i.e. missing or highly reduced reflexes in LM? I fear that these subjects should have been excluded instead?

RESPONSE: We agree with the concern expressed by Reviewer 1 regarding the use of TES as a target muscle for two participants. We use this method to avoid excluding these participants based on an absence of reflex in LM. Our basis for this decision was that our objective was not only to compare the thresholds at L3 and the LM motor responses (objective #1), but to observe muscle strategies and receptive fields encompassing multiple trunk sites and muscles (objective #2). Thus, we needed a way to set the stimulation intensity to being able to stimulate the other trunk sites. We do not think that this method biases our results. As depicted in Fig. 1, the two participants for which TES was used to set NWR threshold had the highest and 5th highest NWR threshold in lying. Excluding these participants would have also affected the results as it would have led to under-estimation of the average NWR threshold and reduce the generalizability of the results. Another possibility would be to consider these two participants as having a “fixed” NWR threshold (e.g., equal to the largest NWR or to 100 mA) to ease the comparisons between groups (only NWR threshold of LM is used). Nevertheless, even when considering that LM was 100 mA for these two participants, it does not change the results (p=0.07). 

Moreover, the latter method does not permit to set a stimulation intensity at which stimulating the other trunk sites. We acknowledge that this method was not optimal but we believe it was the best way to avoid exclusion of participants and adjust stimulation intensity for other sites to allow estimation of motor strategies and receptive fields one of the study objectives.

We added a sentence in the Methods:

p. 6-7; ln 157-159: “This method allows to determine an intensity to stimulate other trunk sites to estimate motor strategies and receptive fields (objective #2).”

4. Please elaborate on how NWR thresholding was made, several studies have investigated the methodology of optimizing the NWR threshold determination (I suggest that the author see the work by Rhudy and France, see also specific comments below).

RESPONSE: The detection of NWR was not automatized based on online analysis of the response to noxious stimulus as suggested by Rhudy and France (5). Those criteria have been established using the nociceptive flexion reflex (NFR) that consists of the recording of the biceps femoris EMG responses to an electrical noxious stimulation of the sural nerve. In contrast, as threshold, we measured the EMG responses of the LM in response to an electrical cutaneous noxious stimulation of the lower back skin i.e., a nociceptive withdrawal reflex (NWR). Other research groups testing NWR threshold of the foot (stimulation of the foot arch) used different criteria than the one proposed by Rhudy and France (e.g., 20 μV for at least 10 ms in the 50- to 150-ms poststimulation (6, 7)). A major issue is that objective automated criteria are difficult to apply to back muscles because of the different characteristics of the EMG signal. A major distinction is that low back muscle EMG is characterised by a low signal-to-noise ratio, differences in muscle mass, electrode positioning, and complexity of back muscle anatomy. Together this means that signal amplitudes differ substantially between individuals and preclude the use of absolute EMG amplitude criteria between different study individuals and groups. Further, the short peripheral nerve conduction time means that it is more likely that the response might be contaminated by stimulus artifact and the presence of stimulus artifact would have impeded to use of any automated criteria during thresholding.

 It is also important to highlight that the gold standard at which the objective criteria from the study of Rhudy and France (2007) were compared was a visual identification of an absence/presence of NWR. We used this technique in the current study. About our thresholding methods, the intensity was increased by small increments until LM responses were observed over “noise” (at rest) or EMG background (sitting - active) in the window 40-200 ms poststimulation. The intensity was adjusted (increased or reduced) until LM responses were observed in 50% of trials Then, the latter steps were repeated until the intensity was confirmed and this intensity was considered the NWR threshold.

We added these detail in the manuscript. 

Methods:

p. 6; Ln 150-156: “To determine the NWR threshold, current intensity began at 2 mA with the probe placed at L3. This was increased by small increments until a response was evoked in the LM EMG recording within the 40-200-ms window post-stimulation (8). Although this was generally characterized by excitation, inhibition was sometimes observed as the earliest response when the participant was sitting. At this point, the stimulus intensity was slightly increased/decreased until LM responses were evoked in 50% of the stimulations (3 out of 6). Then, the latter steps were repeated until the intensity was confirmed and this intensity was considered the NWR threshold.”

5. Please specify how many stimuli/reflexes have been omitted to stimulus artefacts.

RESPONSE: For LM with stimulation at S1, 4 participants were removed from the analysis. For TES and LM with stimulation at T12 and L3, 3 participants were omitted. These details are available in the first section of the Results section (p.11; Ln 273-275).

6. The details about the ‘2-branch probe’ stimulation electrode must be clarified, possibly with a figure. How was the electrode fixed to the skin? Is this a similar electrode to that used by (Rukwied, Schmelz et al 2020) i.e. the positioning is handheld? If so, this means that the stimulation site alters with every stimulation at the ‘site’, resulting in slightly different afferents being stimulated and different impedances. Meaning that the stimulation intensity in relation to NWR threshold differs even more. Please clarify.

RESPONSE: The 2-branch probe was handheld - the experimenter applied the electrodes against the skin and maintain it for the duration of the blocks of stimulation. The experimenter carefully checked that the probe remains still and in the same position during the whole block of stimulation. We agree that it may have slightly modified the afferent recruited during the experiment although we believe it is unlikely that the handheld positioning biased the results. Indeed, considering that 6 noxious stimuli were performed at each site, experimenter did not have time to fatigue. We added detail in the methods and discussion. 

Methods:

p. 6; Ln 143-147: “Electrical stimuli were delivered as a constant current pulse train of 5 single 0.2 ms square-wave pulses with a 2-ms inter-pulse interval (Digitimer DS7AH [maximal current 1A], Hertfordshire, United Kingdom) using a 2-branch handheld ‘probe’ (gold electrodes, 10 mm inter-electrode distance, ~0.8 mm2 contact). The experimenter positioned the probe on the participant’s skin and carefully monitored its position and the applied pressure during each block of stimulation.”

Discussion:

p. 20; Ln 414-415: “Fourth, the electrical stimulation was done using a handheld probe that may have slightly influenced the afferent recruited during the stimulation.”

7. The manuscript should be expanded with examples of EMG traces from both CBLP and control. And possibly also with each subgroup of CBLP. I am curious to see / confirm the traces and whether there is background activity prior to electrical stimuli? I suggest the authors to illustrate the traces from the pre-stimulus period to end of the late response - possibly also including the electrical artefact, as a reader can differentiate that to actual EMG activity.

RESPONSE: A new Figure has been added to address this comment – New Fig. 2.

8. I believe that the supplementary material should be included as data in the main manuscript. Because this data is in my view better suited to answer the original hypothesis of the study (regarding NWR thresholds, and thus, the NWR responses as well, in my view). The subdivision of the LBP individuals is not directly related to the main hypothesis, but more the secondary hypothesis (see other comments).

RESPONSE: We agree with Reviewer 1. We substantially modified the manuscript to include the supplementary material in the Results section. Tables and Figures have been also modified to present the results from the “Group” analyses (CLBP vs. CTL). We kept some parts of the Subgroup analyses in the main manuscript in cases where this explained observations regarding the differences between CTL and CLBP. All results from the subgroup analyses have been moved to Supplementary materials (Figures, Tables and Results section).

9. The first part of the introduction and evidence that individuals with CBLP move differently should be clarified, so that it is clear how the trunk movement during CLBP is affected.

RESPONSE: We added some detail accordingly.

Introduction:

p. 3; Ln 60-63: “For example, studies have reported increased activation of superficial erector spinae (1) and abdominal (2) muscles during forward bending, and delayed activation of deep multifidus muscles during postural tasks (3) and walking (4) in participants with CLBP compared to painfree controls.”

10. The manuscript is not very clear as how the stimulation intensity was adjusted over the course of the experiment. Initially an intensity of 2x NWR threshold was used, but this was then adjusted based on reported pain, this description is not sufficiently accurate, and does allow for reproducing the methods. How often was this adjusted, and to what degree? Was it done in all subjects?

RESPONSE: We added detail in the methods.

p. 7; Ln 163-171: “NWR threshold was only tested at L3 using LM motor responses. Considering the high number of painful stimuli and the time taken to find NWR threshold, it was not feasible to identify thresholds individually at each site and in both positions. To enable comparisons between sites, reported pain at L3 spinous process at 2 times NWR threshold was matched between sites by adjustment of stimulation intensity. At sites other than L3, participants rated the pain intensity following the first stimulus. If pain intensity differed from the target by 2 or more out of 10 (the minimal clinically important difference for NRS (35)), the intensity was adjusted, and additional series of stimuli were performed until the pain intensity was reported at the same intensity as that induced by stimulation at L3. This technique was used to maintain the amplitude of reported pain at each site.”

11. Why was the reflex magnitude reported as normalized to MVC?

RESPONSE: Particularly for the back muscles where EMG amplitude depends on many factors including subcutaneous fat, muscle fibre orientation, and so on, normalisation using MVC is recommended to allow comparisons between-group and between-muscles analyses. This is the preferred methods as reported by the Consensus for experimental design in electromyography (CEDE) project (12). For evoked responses studies, M-wave technique is usually recommended but this is not possible for trunk muscles because of the complexity to stimulate their motor axons. Using an MVC allows to perform these comparisons.

We added this detail in the method section:

p. 9; Ln 211-213: “The MVC normalisation technique is recommenced by the Consensus for Experimental Design in Electromyography (CEDE) project to allow between-group and between-muscle comparisons (38).”

12. Please add more info about EMG electrode types, such as model name.

RESPONSE: We added this detail in the Methods.

13. Why was the EMG amplification not adjusted to each individual? Individual anatomical difference, e.g. adipose layers etc may greatly affect the EMG signal and thus the need for amplification.

RESPONSE: The EMG amplification was maintained stable but signals were normalised to control for the effects identified by the reviewer.

14. I find the results section(s) regarding the receptive fields to be unintuitive to understand. Fig 2 and 3 contain vast amounts of data but it is difficult to elude the details of the receptive fields based on these figures. If the authors wish to describe the receptive fields of different muscles (and the differences between CTL and CLBP), why not actually map the receptive fields based on the data (similar to Massé-Alarie, EJN, 2019) and then quantify the sizes and other characteristics of the RRF?

RESPONSE: We acknowledge that it may be difficult to understand some sections of the results based on the terms motor strategies and receptive fields. Motor strategies and receptive fields differences were inferred only when significant Group x Site was detected (or Subgroup x Site for additional analyses). When an interaction was detected, any between-group differences in the activation of a given muscles based on post-hoc comparisons was considered as difference in motor strategies. For receptive fields, within-group comparisons were used, that is when there were differences in the activation of a given muscle across sites. The Figures 3 and 4 are designed to visualise receptive fields for each muscle (i.e., each muscle activation can be compared across sites in each panel). Also, the Table 3 presents the statistical difference between sites for a given muscle (i.e. receptive field – within-group comparison) separately for the two groups. One issue with the term receptive field is that we only stimulated few sites, thus, that makes its quantification difficult in contrast to quantification of foot NWR receptive field (13, 14). 

In our previous work, we did not map the receptive field but rather the motor strategy (15). The figure was only a visual representation of the different muscles activated while stimulating each site separately. No quantitative variables were extracted from that analysis.

We clarified these aspects in Methods and Results sections:

Methods:

p. 10; Ln 240-244: “We interpreted receptive field and motor strategy differences only in presence of significant Group x Site interactions. Differences in motor strategy was considered in presence of significant post-hoc comparisons between groups (i.e. between-group comparisons) for each site and muscle. Differences in receptive field was considered in presence of significant post-hoc comparisons in reflex amplitude/occurrence between sites (i.e. within-group comparison), for each muscle and group.”

Results:

p. 15; Ln 320-321“Motor strategies represent between-group differences in occurrence/amplitude of muscle responses after stimulation of a given site by post-hoc analyses of significant interactions.”

p. 15; Ln 336-337: “Receptive fields were considered by within-group comparisons of occurrence/amplitude of responses of a given muscle across stimulation sites independently for each group.”

15. Please add statistic test to paratheses where p-values are described, both in the main text and figures/tables captions.

RESPONSE: The Wald χ2 and p-values of the Group x Site interaction and main effects of Group are displayed in the Results section and in Table 3. We added the p-values of the post-hoc comparisons in the manuscript and Figures.

16. In first part of the discussion the conclusions regarding the two CBLP subgroups should be soften up, due to the low group sizes.

RESPONSE: We agree and added sentences in Discussion and Conclusion to soften the interpretation. In addition, we now interpret results from the whole CLBP group.

Discussion

p. 19; Ln 398-399: “Due to the small sample size, non-significant results can be the result of insufficient power to detect differences and should be interpreted cautiously.”

Conclusion:

p.24; Ln 510-511: “Due to the small sample size, replication of our results with a larger sample size would be beneficial.”

17. If compliant with Journal rules, I suggest changing the short title to “Different nociceptive withdrawal reflex in low back pain” I believe that adding ‘nociceptive’ clarifies the title.

RESPONSE: We modified accordingly.

18. There are several minor typo errors in the manuscript such as unneeded capital letters or missing capital letter (for example in figures), missing spaces between text and paratheses for reference etc. Additionally several sentences lack clarity, making it difficult to understand the meaning. Please proof-read the manuscript carefully.

RESPONSE: The manuscript has been revised carefully to improve its clarity.

Specific comments.

Abstract L 48 Please check sentence construction in ‘Two CLBP subgroups were identified by different in NWR threshold:’

RESPONSE: This sentence was removed after substantial revision of the article.

Please be specific/consistent when using abbreviations. E.g. CLBP and LBP, I believe they are used too interchangeably to refer to pain sufferers. Would it better to simply use one?

RESPONSE: We used LBP when we refer to the whole population of individuals suffering low back pain, not specifically chronic. To reduce the confusion, we did not use the acronym LBP but rather write “low back pain” when it was not specific to chronic low back pain.

L87-92 I do not believe that this secondary hypothesis is very well underlined, nor do I believe it is directly related to the subdivision of CBLP subjects (see later comments).

RESPONSE: We agree with Reviewer 1. The revised manuscript now presents results related to these objectives (comparisons between CLBP and CTL).

L133-137 what was the order of the analog EMG filter (BP 5Hz - 1kHz)? I assume it is ringing from these filters, which causes problems with the stimulation artefacts? I wonder if this could have been solved by adjusted the filter order, thus reducing the ringing.

RESPONSE: The NL125 filter is an analogue 2 pole resistor-capacitor that does not incorporate any additional features such as Butterworth. Thus, there is no filter order (information coming from Digitimer). 

P7 L147. I fear that NWR thresholding is not very systematic. The authors states ‘increase until a response was evoked’ what was the criteria for evoked reflex? I refer the authors but the works by Rhudy and France (2007/2009) etc on how to determine the threshold of the NWR. Additionally, in what steps were the intensity increased. There were no ‘staircase’ procedure or similar? (see also general comments above)

RESPONSE: We refer Reviewer 1 to our detailed response to general comments #4.

Fig 2B, please ensure y-label starts with capital letter.

RESPONSE: We modified accordingly.

L 143-4 I do not see the relevance or meaning of the sentence ‘Stimulation with 0.2-ms pulse width induces pain of 3-7/10 with stimuli of ~150mA to the quadriceps muscles (31).’ The electrical current is not easily translatable across different body regions, and different electrode configurations etc. I suggest clarifying or deleting.

RESPONSE: This sentence was removed. 

P6 L 144. The phrase “This was perceived by the participant as a single noxious stimulus.” should have the word noxious removed.

RESPONSE: We removed the word “noxious”.

L203-5 Why was this criterion for reflex detection used? What is the rationale for using this? Again I refer the authors to comprehensive work by Rhudy and France.

RESPONSE: As discussed in the response to general comment #4, we do not think that the criteria from Rhudy & France are suitable to apply to the reflex evoked in our study considering the protocol is quite different (NFR vs. NWR; trunk muscles vs. biceps brachii; need for post hoc filtering to limit the impact of the artifact on EMG amplitude, etc.). Thus, we used criteria that has been shown to be sensitive to detect EMG onset of trunk muscles (17). 

L208 The division of the LBP subject was made based on the data CTL and the 95 % CI. What is the rationale behind this, and how does this fit with the initial hypothesis? What other analysis were made to investigate this division?

RESPONSE: The paper is now centered around the “group” comparisons that better fit the study objectives. 

L217-219 While I appreciate the authors interest in showing the level of variation in LBP. I urge the author to be careful with the conclusion based on the two subgroups of the LBP group because of the very small sample sizes. I am in doubt of whether this rather strong division is a general phenomenon in CLBP.

RESPONSE: The conclusion was modified to be more careful and we now present results from the comparisons between groups (CLBP vs. CTL) in the main manuscript and modified the discussion to interpret findings from these analyses. Additional subgroup analyses were used to highlight which results were completely driven by the high-treshold subgroups and is now available as supplementary materials.

L253 Why is Site and Position written in capitals?

RESPONSE: Capital letters at the beginning of the word were to identify that position and site were factors in the factorial analyses used. We removed the capital letters.

L 258-9 please clarify the sentence ‘…. could not be assessed, but the presence or not of an early response was recorded.’

RESPONSE: Although the artifact impeded the measure of early responses amplitude (due to signal derivation), it was still possible to identify the presence of an EMG response. We clarified this sentence.

p. 11; Ln 280-282: “Although the amplitude of the early response could not be assessed because of the presence of EMG artifact, it was possible to determine if an early response was present.”

L372, delete ’ 6-fold’ this is not accurate for all cases.

RESPONSE: 6-fold was removed.

L379-380 I strongly agree. Also why the conclusion regarding the subgroup must be soften up.

RESPONSE: We agree and also added a sentence in the conclusion of the study. 

p. 24; Ln 510-511: “Due to the small sample size, replication of our results with a larger sample size would be beneficial.”

L391-392 this sentence is very hard to understand. In what way is the amplitude results the same as the number of occurrences?

RESPONSE: This sentence was removed since the information was redundant to the previous sentence. We added some detail to clarify the sentence.

L401 ‘This concurs with the response of the High-threshold group.’ I am not sure I agree with the authors here. In many of the studies being reference in sentences prior, common is that those subjects, where no reflexes could be elicited were often excluded. However, in the current study, subjects where reflexes were difficult to elicit, stimuli were merely given to another site to evoke reflexes from there. As mentioned above I am very doubtful of whether these subjects should have been included in the analysis.

RESPONSE: We respectfully disagree with Reviewer 1. Excluding these subjects without considering them in the results also introduce some biases. Indeed, studies observed a lower NWR threshold in LBP compared to painfree controls. However, the participants suffering LBP without NWR were excluded of the analyses and authors did not “set” any value for their threshold. If 30% of participants without NWR were ascribed a NWR threshold (e.g., a fixed value of 50 mA), it is possible that the difference in threshold would have been drastically reduced. Thus, we argue that the methods used in previous studies are likely to underestimate the real value of the NWR threshold in LBP. 

L403 ‘In humans, no study has tested NWR within the clinical pain area. ’ this sentence is not correct. The authors reference several NWR studies made in chronic pain patients.

RESPONSE: We agree with Reviewer 1, the studies we referenced to included clinical pain population. However, they tested the NWR of the foot or the nociceptive flexion reflex (NFR) i.e., not in the clinical area of pain. 

L422-423 the authors suggest that the high-thr have sensitization, creating this ‘overaction’. However, in case of CS that would mean lower NWR threshold (rather than higher threshold). Instead I believe this overaction of the abdominal muscles are due to the stimulation intensity being insufficiently calibrated to each stimulation site and the high-thr group using (too) high stimulation intensity.

RESPONSE: We agree that it is possible that the higher responses in abdominal muscles following stimulation of the rib was due to the lack of calibration based on NWR threshold. However, we adapted the stimulation intensity across sites based on pain intensity. There was no difference in pain intensity between sites and between groups. We believe that a similar level of pain allows comparison of motor responses linked to noxious stimuli. Pain is an output from the brain and defined by the International Association for Study of Pain (IASP) as “An unpleasant sensory and emotional experience associated with, or resembling that associated with, actual or potential tissue damage” (18). It is an integration of genetic, contextual, psychological and nociceptive information at the time of the experiment. For our experiment, pain is probably strongly driven by nociceptive afferent to the central nervous system supported by the linear relation between amplitude of the NFR and pain intensity (e.g., in (19)). Similar pain intensity may signify that the perceived threat of the noxious stimuli is similar between sites and groups, and thus, could be compared. 

It is interesting to observe that although the high threshold participants had very large response in abdominal muscles but no response in LM following S1 stimulation despite very large intensity of stimulation was used for both sites. Considering the pain intensity reported was the same (and very large), it is unlikely nociceptive afferent were not recruited.

The ‘Sensitization in CLBP’ section of the discussion is not very well structured and lacks focus, making it difficult to follow. Perhaps considering shortening, as I am not sure all content is relevant for the discussion of the current results.

RESPONSE: This section was modified, please see amended text in the document.

Fig 1. Please add details on statistic. Was the data in 1A normally distributed?

RESPONSE: Data in Fig. 1 was not normally distributed, a non-parametric test was applied. There was no difference between CLBP and CTL groups (p=0.06). We did not compute statistical analyses on subgroups since they were divided based on NWR threshold. 

Fig 2B + 3B. Occurrence with large capital letter, but perhaps the word ‘Frequency’ is more appropriate? Please add units. If this is occurrence how can the number be less than 1 (this has not been explained in neither methods or captions). I suggest to report this in percentages. Add the definition of early response in the figure caption.

RESPONSE: We modified the figures accordingly and added a description of the measurement of frequency of occurrence (ratio), in the methods and captions.

Methods:

p. 9; ln 218-220: “We calculated the frequency of occurrence as a ratio between the number of identifiable early responses divided by the total number of stimulations.”

Generally Fig 2 and 3 are intended to allow comparison across stimulation sites and muscles. However, the y-scaling is not identical for any plots, greatly limiting the comparison across parameters. I strongly suggest the authors to re-consider this, especially as the main text tries to compare across sites and muscles.

RESPONSE: We agree that it may be difficult to compare the amplitude/occurrence of activation of muscles after a given site in Fig. 3 and 4. However, we decided to optimize the y-scaling for each plot to facilitate the comparison of activation/occurrence for each muscle across sites (receptive field). We believe that the figures fit how the results are presented and allow comparison between groups for a same muscle after stimulation of a given site. Further, we argue that the way the data are presented better represents the statistical models that we used (Group x Site). Using the same scaling across plots would make the comparisons difficult across sites for a given muscle considering the activation differences (e.g., RA is usually less activated than OE). 

Please add label for the x-axis for Fig 1-4, such as stimulation site. I recommend only doing so for the bottom row in each figure.

RESPONSE: We added labels as requested.

 

Response to Reviewer #2: 

This exploratory study set out to determine whether the organisation and excitability of the trunk nociceptive withdrawal reflexes (NWR) are modified in chronic low back pain (CLBP). The authors hypothesised that individuals with CLBP would have modified NWR pattern and lower NWR thresholds. Electrical stimulation was delivered over the spine at different vertebral levels and over the 8th rib in order to elicit NWR in 12 participants with CLBP and data compared with that collected previously from healthy participants, by the same authors, using an identical protocol. Surface electromyographic (EMG) activity was recorded from 5 trunk muscles and occurrence and sizes of responses were determined. The results showed that 2 CLBP subgroups were identified by differences in NWR thresholds, high and low. The responses in abdominal muscles were larger in those with high thresholds. Occurrence of response in multifidus were lower in CLBP groups than in controls. The authors conclude that the results suggest modified networks in spinal networks control trunk muscles could explain differences in motor control in those with CLBP.

The study is well written and experiments seem diligently performed. The results are novel and suggest differences in organisation of motor spinal networks in CLPB, which add to the evidence of changes in supraspinal areas. Caution is advised though as the n numbers were small in both the previously published control data and the current CLBP group.

RESPONSE: We thank Reviewer 2 for their positive comments.

1. I think it would be good to include some brief discussion of the idea that any changes in NWR thresholds and amplitudes might be related to the changes known to exist in supraspinal regions (e.g. motor cortex) in CLBP. In healthy participants, NWR induced by higher intensity stimulation appears to activate top-down analgesic mechanisms, which can be reduced with tDCS of the motor cortex (Hughes et al., 2019). NWR induced in those with ongoing CLBP are likely influenced by the co-existing changes within brain regions. There is only brief mention of the interplay between spinal and supraspinal circuits (lines 462-465) essentially stating that changes in spinal networks contribute to alterations in spine control along with changes in supraspinal areas.

RESPONSE: We agree with Reviewer 2 and added some discussion about these points.

Discussion

p. 23; ln 499-501: “Also, it is suggested that M1 may contribute to top-down control of nociception and pain processing (see references (64, 65)), and it could in part explain differences observed in NWR in CLBP.”

Minor comments:

Line 65. I wonder if the sentence “Assessment of spinal motor networks that control back muscles is challenging” should be altered to say “trunk” instead of “back”, particularly since the preceding sentences in this paragraph related to the trunk, rather Line 72 – Change “controlled at the spinal cord” to “controlled at the level of the spinal cord”.

RESPONSE: We modified this sentence accordingly.

Line 73 – Change “and influenced by supraspinal control” to “and is influenced by supraspinal control”

RESPONSE: We modified the sentence.

Lines 76-77 – “in a manner consistent with its biomechanical function”. Does reference 21 apply to this too? If so, maybe it can be added at the end of this sentence?

RESPONSE: The reference was moved at the end of the sentence.

Lines 204-205. “when the EMG amplitude exceeded the mean of the pre-stimulus EMG activity (100-ms window) by 1 standard deviation for 50 ms, and this was confirmed visually (35).” Although this is referenced, this seems an unusually low level to exceed in order to state that a response had occurred. Is this a mistake and it should be exceeding 2SDs?

RESPONSE: No, this is not a mistake. We used this technique to be sensitive enough to avoid missing real increase in EMG. For example, it has been shown that the use of 2 and 3 SD may increase the number of missing values in the selection of onset (17). Considering that the signal-to-noise ratio may be low for some participants, we decided to use a more sensitive technique and adjust when necessary. Moreover, all selection of onsets were visually confirmed or adjusted by an experienced evaluator that is the gold standard. 

Lines 428-431. This is quite speculative, is there any evidence that NWR can be intentionally suppressed?

RESPONSE: We agree with Reviewer 2 that this sentence is speculative. We did not suggest that an individual will consciously suppress but rather that the central nervous system may want to suppress this reflex that could potentially increase pain. 

p. 21-22; Ln 460-461: “Although speculative, the central nervous system might suppress/reduce the NWR to avoid pain expected to be induced by movement.”

Reference:

Hughes, S., Grimsey, S., Strutton, P.H. (2018). Primary motor cortex transcranial direct current stimulation modulates temporal summation of the nociceptive withdrawal reflex in healthy subjects. Pain Medicine. 20(6):1156-1165). doi: 10.1093/pm/pny200.

Bibliography

1. Childs JD, Piva SR, Fritz JM. Responsiveness of the numeric pain rating scale in patients with low back pain. Spine (Phila Pa 1976). 2005;30(11):1331-4.

2. Masse-Alarie H, Angarita-Fonseca A, Lacasse A, Page MG, Tetreault P, Fortin M, et al. Low back pain definitions: effect on patient inclusion and clinical profiles. Pain Rep. 2022;7(2):e997.

3. Kjaer P, Bendix T, Sorensen JS, Korsholm L, Leboeuf-Yde C. Are MRI-defined fat infiltrations in the multifidus muscles associated with low back pain? BMC Med. 2007;5:2.

4. MacDonald D, Moseley GL, Hodges PW. Why do some patients keep hurting their back? Evidence of ongoing back muscle dysfunction during remission from recurrent back pain. Pain. 2009;142(3):183-8.

5. Rhudy JL, France CR. Defining the nociceptive flexion reflex (NFR) threshold in human participants: a comparison of different scoring criteria. Pain. 2007;128(3):244-53.

6. Curatolo M, Muller M, Ashraf A, Neziri AY, Streitberger K, Andersen OK, et al. Pain hypersensitivity and spinal nociceptive hypersensitivity in chronic pain: prevalence and associated factors. Pain. 2015;156(11):2373-82.

7. Biurrun Manresa JA, Neziri AY, Curatolo M, Arendt-Nielsen L, Andersen OK. Reflex receptive fields are enlarged in patients with musculoskeletal low back and neck pain. Pain. 2013;154(8):1318-24.

8. Andersen OK, Sonnenborg FA, Arendt-Nielsen L. Modular organization of human leg withdrawal reflexes elicited by electrical stimulation of the foot sole. Muscle & nerve. 1999;22(11):1520-30.

9. Geisser ME, Ranavaya M, Haig AJ, Roth RS, Zucker R, Ambroz C, et al. A meta-analytic review of surface electromyography among persons with low back pain and normal, healthy controls. J Pain. 2005;6(11):711-26.

10. Massé-Alarie H, Beaulieu LD, Preuss R, Schneider C. Influence of chronic low back pain and fear of movement on the activation of the transversely oriented abdominal muscles during forward bending. J Electromyogr Kinesiol. 2016;27:87-94.

11. Smith JA, Kulig K. Altered multifidus recruitment during walking in young asymptomatic individuals with a history of low back pain. journal of orthopaedic & sports physical therapy. 2016;46(5):365-74.

12. Besomi M, Hodges PW, Clancy EA, Van Dieen J, Hug F, Lowery M, et al. Consensus for experimental design in electromyography (CEDE) project: Amplitude normalization matrix. J Electromyogr Kinesiol. 2020;53:102438.

13. Andersen OK, Sonnenborg F, Matjacic Z, Arendt-Nielsen L. Foot-sole reflex receptive fields for human withdrawal reflexes in symmetrical standing position. Exp Brain Res. 2003;152(4):434-43.

14. Andersen OK, Sonnenborg FA, Arendt-Nielsen L. Reflex receptive fields for human withdrawal reflexes elicited by non-painful and painful electrical stimulation of the foot sole. Clin Neurophysiol. 2001;112(4):641-9.

15. Masse-Alarie H, Salomoni SE, Hodges PW. The nociceptive withdrawal reflex of the trunk is organized with unique muscle receptive fields and motor strategies. The European journal of neuroscience. 2019.

16. Gallina A, Abboud J, Blouin JS. A task-relevant experimental pain model to target motor adaptation. J Physiol. 2021;599(9):2401-17.

17. Hodges PW, Bui BH. A comparison of computer-based methods for the determination of onset of muscle contraction using electromyography. Electroencephalogr Clin Neurophysiol. 1996;101(6):511-9.

18. Pain IAfSo. Terminology 2021 [cited 2022. Available from: https://www.iasp-pain.org/resources/terminology/.

19. de Willer JC. Comparative study of perceived pain and nociceptive flexion reflex in man. Pain. 1977;3(1):69-80.

---

## [Decision Letter · Decision Letter 1]

27 Jan 2023

PONE-D-22-26918R1Nociceptive withdrawal reflexes of the trunk muscles in chronic low back painPLOS ONE

Dear Dr. Massé-Alarie,

Thank you for submitting your manuscript to PLOS ONE. After careful consideration, we feel that it has merit but does not fully meet PLOS ONE’s publication criteria as it currently stands. Therefore, we invite you to submit a revised version of the manuscript that addresses the points raised during the review process. The revised version was relatively well received by the Reviewers. However, there are still some lingering issues that will require your attention. In particular, Reviewer #1 has still questions about certain methodological aspects (threshold determinations) and some suggestions to improve the manuscript. In that regard, although I agree with most of the Reviewer's suggestions, I must disagree with the one regarding the interpretation of near-significant results(Table 2, p=0,09). While It might be tempting to discuss such results as 'significant' trends, we should avoid such interpretation and just report that no differences existed when the analysis failed to reach significance.

We look forward to receiving your revised manuscript.

Kind regards,

François Tremblay, PhD

Academic Editor

PLOS ONE

Journal Requirements:

Reviewers' comments:

Reviewer's Responses to Questions

**Comments to the Author**

1. If the authors have adequately addressed your comments raised in a previous round of review and you feel that this manuscript is now acceptable for publication, you may indicate that here to bypass the “Comments to the Author” section, enter your conflict of interest statement in the “Confidential to Editor” section, and submit your "Accept" recommendation.

Reviewer #1: (No Response)

Reviewer #2: All comments have been addressed

2. Is the manuscript technically sound, and do the data support the conclusions?

Reviewer #1: Partly

Reviewer #2: Yes

3. Has the statistical analysis been performed appropriately and rigorously? 

Reviewer #1: Yes

Reviewer #2: Yes

4. Have the authors made all data underlying the findings in their manuscript fully available?

Reviewer #1: Yes

Reviewer #2: Yes

5. Is the manuscript presented in an intelligible fashion and written in standard English?

Reviewer #1: Yes

Reviewer #2: Yes

6. Review Comments to the Author

Reviewer #1: I thank the authors for critically revising the manuscript. I think the manuscript has significantly improved with this last revision. My main concerns regarding reflex thresholding and elicitation have been further elaborated. I, however, still have a few comments which should be addressed.

Q: Response 2: regarding NWR threshold determination, while it may be time consuming to map the NWR threshold for all stimulation sites, I do not believe that the thresholding is as painful as the reflex recording itself, which uses 2x higher intensities, and will thus evoked greater discomfort and pain. Thus, in L 161 “painful stimuli and” should be removed.

Q: The sentence “To enable comparisons between sites, reported pain at L3 spinous process at 2 times the NWR threshold was matched between sites by adjustment of stimulation intensity” is not very clear to me. Because I believe it is not the NWR threshold that is matched (it is not recorded for most sites), but the stimulation intensity to elicit the reflexes was matched based on the perceived pain? I suggest a sentence similar to “To enable comparisons between sites, the stimulation intensity at other sites was adjusted to elicit the same degree of pain as stimulation at 2x NWR threshold at L3.”

Q: In Table2 the authors report no significant main effect but still a p value of 0.09, indicating there is a quite a bit of variation in the perceived pain. Thus the sentence “It is noteworthy that the pain elicited by the electrical stimulus was not reported to differ between CLBP and CTL despite the use of larger intensity of stimulation in CLBP. ” should perhaps be soften up slightly to indicate that T2 indicate that the data are close to showing significant differences.

Q: Response 3: The use of ‘objective #2’ is not very clear, if this is has not been used previously in the MS. In the introduction the authors call them aims, consider using the same wording.

Q: Regarding response 4: What was the definition of “response was evoked in the LM EMG”. What it based on an amplitude, peak2peak amplitude above X, or z-score or similar?

Q: Response 5: please clarify that the number excluded (n) is referring to entire subjects and not just single reflexes.

Q: p. 6; Ln 148-156: Please be more specific as to what is ‘small increments’. Normally this is either a fixed step size or an adaptive step which changes with each deflection point (reflex present / not present).

Q: The sentence in the discussion “Third, we were unable to measure the early response amplitude from some participants due to contamination of the early response by electrical.” Should be corrected, I believe “stimulation artefacts “ might be missing?

Q: p. 21, Ln 449-455: “It is important to note we did not assess Rib NWR threshold individually.” I suggest to underline to include a sentence similar to “NWR threshold was not individually assessed individually at each stimulation site.”

Q: Figure 2. thank you for including this figure. I think it is crucial to see this. Please indicate onset of stimulus. Please clarify what the grey traces represent. I suggest using an x-axis rather than a scaling bar, timing is a critical parameter to understand in relation to the NWR, especially when doing an early and late component analysis. Could you indicate the ‘response windows’? I believe this was 40-80 ms for the early component? And in relation to this, I notice that you have significant ringing after the stimulation artefact, it is important to ensure that these did not cover any of the analyzed reflex windows. Because based on fig 2 the ringing is still present beoynd 50 ms? This is hugely problematic and will effect the analysis of the early components. If ringing is present to this degree, I fear the analysis of the early component is invalid.

Q: Fig 2 legend: I do not understand the sentence “Note the large artifact that may influence EMG signal but the possibility to identify reflex occurrences.”

Reviewer #2: Thanks for addressing my comments. One small comment - in Table 1 there is a superscript 2 in the column with P values in the Gender row, but this is not mentioned in the legend, whereas the superscript 1 is (1Fisher Exact test used.)

7. PLOS authors have the option to publish the peer review history of their article (what does this mean?). If published, this will include your full peer review and any attached files.

Reviewer #1: No

Reviewer #2: No

---

## [Author Response · Author response to Decision Letter 1]

17 Apr 2023

PONE-D-22-26918R1

Nociceptive withdrawal reflexes of the trunk muscles in chronic low back pain

Response to Reviewer #1:

I thank the authors for critically revising the manuscript. I think the manuscript has significantly improved with this last revision. My main concerns regarding reflex thresholding and elicitation have been further elaborated. I, however, still have a few comments which should be addressed.

RESPONSE: We are pleased that Reviewer 1 appreciate the effort done to revise the manuscript.

Q: Response 2: regarding NWR threshold determination, while it may be time consuming to map the NWR threshold for all stimulation sites, I do not believe that the thresholding is as painful as the reflex recording itself, which uses 2x higher intensities, and will thus evoked greater discomfort and pain. Thus, in L 161 “painful stimuli and” should be removed.

RESPONSE: We remove “painful stimuli and” from the manuscript.

Q: The sentence “To enable comparisons between sites, reported pain at L3 spinous process at 2 times the NWR threshold was matched between sites by adjustment of stimulation intensity” is not very clear to me. Because I believe it is not the NWR threshold that is matched (it is not recorded for most sites), but the stimulation intensity to elicit the reflexes was matched based on the perceived pain? I suggest a sentence similar to “To enable comparisons between sites, the stimulation intensity at other sites was adjusted to elicit the same degree of pain as stimulation at 2x NWR threshold at L3.”

RESPONSE: We agree with Reviewer 1 and modified the manuscript accordingly.

Q: In Table2 the authors report no significant main effect but still a p value of 0.09, indicating there is a quite a bit of variation in the perceived pain. Thus the sentence “It is noteworthy that the pain elicited by the electrical stimulus was not reported to differ between CLBP and CTL despite the use of larger intensity of stimulation in CLBP. ” should perhaps be soften up slightly to indicate that T2 indicate that the data are close to showing significant differences.

RESPONSE: In line with the Editor’s recommendation, we did not modify the manuscript.

“In that regard, although I agree with most of the Reviewer's suggestions, I must disagree with the one regarding the interpretation of near-significant results(Table 2, p=0,09). While It might be tempting to discuss such results as 'significant' trends, we should avoid such interpretation and just report that no differences existed when the analysis failed to reach significance.”

Q: Response 3: The use of ‘objective #2’ is not very clear, if this is has not been used previously in the MS. In the introduction the authors call them aims, consider using the same wording.

RESPONSE: We change the “objective” for “aim”.

Q: Regarding response 4: What was the definition of “response was evoked in the LM EMG”. What it based on an amplitude, peak2peak amplitude above X, or z-score or similar?

RESPONSE: Visual identification of the response was used. Visual identification of the reflex response was the gold standard at which the objective criteria from the study of Rhudy and France (2007) were compared to determine the absence/presence of NWR. We did not use objective criteria for reasons named in the previous review: (e.g. presence of artefact in the unfiltered EMG signals, low signal-to-noise ratio for EMG of low back muscles, differences in muscle mass, electrode positioning, and complexity of back muscle anatomy). We added a sentence in the Methods to clarify.

Methods:

p. 6; Ln 147-153: “To determine the NWR threshold, the current intensity began at 2 mA with the probe placed at L3. This was increased by 2-mA increments until a response was evoked in the LM EMG recording within the 40-200-ms window post-stimulation [35]. In the case of participants with high threshold, 5-mA increment was used. Although this was generally characterized by excitation, inhibition was sometimes observed as the earliest response when the participant was sitting. Motor response was identified visually when EMG clearly increases/decreases in comparisons to the background EMG signal (i.e. prior to stimulation). Visual identification from an expert is the gold standard to identify evoked responses [36].

Q: Response 5: please clarify that the number excluded (n) is referring to entire subjects and not just single reflexes.

RESPONSE: We refer to participants. However, this section has been removed since we removed the early window analysis from the manuscript.

Q: p. 6; Ln 148-156: Please be more specific as to what is ‘small increments’. Normally this is either a fixed step size or an adaptive step which changes with each deflection point (reflex present / not present).

RESPONSE: Usually, 2 mA was used as small increments. However, in the case of participants with high NWR threshold, 5 mA was used. This was added in the Mehods.

Methods

p.6; ln 148-150: “This was increased by 2-mA increments until a response was evoked in the LM EMG recording within the 40-200-ms window post-stimulation [35]. In the case of participants with high threshold, 5-mA increment was used.”

Q: The sentence in the discussion “Third, we were unable to measure the early response amplitude from some participants due to contamination of the early response by electrical.” Should be corrected, I believe “stimulation artefacts “ might be missing?

RESPONSE: We added “stimulation artifact” accordingly.

Q: p. 21, Ln 449-455: “It is important to note we did not assess Rib NWR threshold individually.” I suggest to underline to include a sentence similar to “NWR threshold was not individually assessed individually at each stimulation site.”

RESPONSE: We modified as suggested by Reviewer 1.

Q: Figure 2. thank you for including this figure. I think it is crucial to see this. Please indicate onset of stimulus. Please clarify what the grey traces represent. I suggest using an x-axis rather than a scaling bar, timing is a critical parameter to understand in relation to the NWR, especially when doing an early and late component analysis. Could you indicate the ‘response windows’? I believe this was 40-80 ms for the early component? And in relation to this, I notice that you have significant ringing after the stimulation artefact, it is important to ensure that these did not cover any of the analyzed reflex windows. Because based on fig 2 the ringing is still present beoynd 50 ms? This is hugely problematic and will effect the analysis of the early components. If ringing is present to this degree, I fear the analysis of the early component is invalid.

RESPONSE: Reviewer 1 is right if the artifact is present in the early window, it invalids the amplitude of the response. Fig 2 presents EMG signals from the muscles closest to the stimulation site (LM from S1), then, the impact of the artifact is larger for these muscles. Although we took care to remove all signals that were impacted by the stimulus artifact, we decided to remove the analysis of the amplitude of the early component, considering that some analyses were computed with very few participants, especially for participants with CLBP for which we use high current intensity. However, we think this is reasonable to keep our analysis of the occurrence of reflex since it was possible to identify them despite the stimulation artifact. The manuscript and figures were modified accordingly.

Q: Fig 2 legend: I do not understand the sentence “Note the large artifact that may influence EMG signal but the possibility to identify reflex occurrences.”

RESPONSE: The Fig 2 legend was modified. The meaning of the ble and grey traces were added.

Fig 2 legend:

Fig 2 Examples of EMG activation of LM after S1 stimulation (upper panels) and EO after rib stimulation (lower panels) in controls (left panels) and participants with CLBP (right panels). The blue traces represent the EMG trace of all stimulation trials averaged. The grey traces represent single trials. The grey boxes represent the late windows (80-200 ms). Large artifact influences EMG signal although it did not impede the identification of early reflex occurrences. LM: Lumbar multifidus; OE: obliquus externus abdominis; S1: first spinous process of the sacrum.

Response to Reviewer #2:

Thanks for addressing my comments. One small comment - in Table 1 there is a superscript 2 in the column with P values in the Gender row, but this is not mentioned in the legend, whereas the superscript 1 is (1Fisher Exact test used.)

RESPONSE: This was changed.

---

## [Decision Letter · Decision Letter 2]

24 May 2023

Nociceptive withdrawal reflexes of the trunk muscles in chronic low back pain

PONE-D-22-26918R2

Dear Dr. Massé-Alarie,

We’re pleased to inform you that your manuscript has been judged scientifically suitable for publication and will be formally accepted for publication once it meets all outstanding technical requirements.

Kind regards,

François Tremblay, PhD

Academic Editor

PLOS ONE

Additional Editor Comments (optional):

Reviewers' comments:

Reviewer's Responses to Questions

**Comments to the Author**

1. If the authors have adequately addressed your comments raised in a previous round of review and you feel that this manuscript is now acceptable for publication, you may indicate that here to bypass the “Comments to the Author” section, enter your conflict of interest statement in the “Confidential to Editor” section, and submit your "Accept" recommendation.

Reviewer #2: All comments have been addressed

2. Is the manuscript technically sound, and do the data support the conclusions?

Reviewer #2: Yes

3. Has the statistical analysis been performed appropriately and rigorously? 

Reviewer #2: Yes

4. Have the authors made all data underlying the findings in their manuscript fully available?

Reviewer #2: Yes

5. Is the manuscript presented in an intelligible fashion and written in standard English?

Reviewer #2: Yes

6. Review Comments to the Author

Reviewer #2: Thank you very much indeed for addressing my most recent comments. I have no further suggestions or queries.

7. PLOS authors have the option to publish the peer review history of their article (what does this mean?). If published, this will include your full peer review and any attached files.

Reviewer #2: No

---

## [Editor Report · Acceptance letter]

6 Jun 2023

PONE-D-22-26918R2 

Nociceptive withdrawal reflexes of the trunk muscles in chronic low back pain 

Dear Dr. Massé-Alarie:

I'm pleased to inform you that your manuscript has been deemed suitable for publication in PLOS ONE. Congratulations! Your manuscript is now with our production department. 

Kind regards, 

on behalf of

Dr. François Tremblay 

Academic Editor

PLOS ONE